# A novel membrane complex is required for docking and regulated exocytosis of lysosome-related organelles in *Tetrahymena thermophila*

**Aarthi Kuppannan** [1‡], **Yu-Yang Jiang** [1‡], **Wolfgang Maier** [2], **Chang Liu** [3], **Charles F. Lang** [4], **Chao-Yin Cheng** [1], **Mark C. Field** [5,6], **Minglei Zhao** [3], **Martin Zoltner** [7], **Aaron P. Turkewitz** [1] *

**1** Molecular Genetics and Cell Biology, The University of Chicago, Chicago, Illinois, United State of America, **2** Bio3/Bioinformatics and Molecular Genetics, Faculty of Biology and ZBMZ, Faculty of Medicine, Albert-Ludwigs-University of Freiburg, Freiburg, Germany, **3** Biochemistry and Molecular Biology, The University of Chicago, Chicago, Illinois, United States of America, **4** Committee on Genetics, Genomics, and Systems Biology, The University of Chicago, Chicago, Illinois, United States of America, **5** School of Life Sciences, University of Dundee, Dundee, Scotland, United Kingdom, **6** Institute of Parasitology, Biology Centre, Czech Academy of Sciences, České Budějovice, Czech Republic, **7** Biotechnology and Biomedicine Centre of the Academy of Sciences and Charles University (BIOCEV), Vestec, Czech Republic

‡ These authors share first authorship on this work.
* apturkew@uchicago.edu

**Data Availability Statement:** All proteomics files are available from the ProteomeXchange Consortium via the PRIDE partner repository

## Abstract

In the ciliate *Tetrahymena thermophila*, lysosome-related organelles called mucocysts accumulate at the cell periphery where they secrete their contents in response to extracellular events, a phenomenon called regulated exocytosis. The molecular bases underlying regulated exocytosis have been extensively described in animals but it is not clear whether similar mechanisms exist in ciliates or their sister lineage, the Apicomplexan parasites, which together belong to the ecologically and medically important superphylum Alveolata. Beginning with a *T. thermophila* mutant in mucocyst exocytosis, we used a forward genetic approach to uncover *MDL1* (**M**ucocyst **D**ischarge with a **L**amG domain), a novel gene that is essential for regulated exocytosis of mucocysts. Mdl1p is a 40 kDa membrane glycoprotein that localizes to mucocysts, and specifically to a tip domain that contacts the plasma membrane when the mucocyst is docked. This sub-localization of Mdl1p, which occurs prior to docking, underscores a functional asymmetry in mucocysts that is strikingly similar to that of highly polarized secretory organelles in other Alveolates. A mis-sense mutation in the LamG domain results in mucocysts that dock but only undergo inefficient exocytosis. In contrast, complete knockout of *MDL1* largely prevents mucocyst docking itself. Mdl1p is physically associated with 9 other proteins, all of them novel and largely restricted to Alveolates, and sedimentation analysis supports the idea that they form a large complex. Analysis of three other members of this putative complex, called MDD (for **M**ucocyst **D**ocking and **D**ischarge), shows that they also localize to mucocysts. Negative staining of purified MDD complexes revealed distinct particles with a central channel. Our results uncover a novel

https://www.ebi.ac.uk/pride/archive/ with the dataset identifier PXD028372. All genome sequencing data are available in the NIH/NCBI SRA database https://www.ncbi.nlm.nih.gov/sra under accession number PRJNA817605.

**Funding:** Grant support came from the Wellcome Trust (https://wellcome.org/grant-funding) grant 204697/Z/16/Z to MCF; Ministry of Education, Youth and Sports of the Czech Republic (https://www.msmt.cz/research-and-development-1) project OPVVV/0000759 to MZ; German Federal Ministry of Education and Research grant (https://fundit.fr/en/institutions/german-federal-ministry-education-and-research-bmbf) 031L0101C de.NBI-epi to WM; US NIH Training Grant T32 (https://researchtraining.nih.gov/programs/training-grants/t32) GM007197, supporting CFL; and US NIH R01 (https://grants.nih.gov/grants/funding/r01.htm) GM105783 to APT. The funders had no role in study design, data collection and analysis, decision to publish, or preparation of the manuscript.

**Competing interests:** The authors have declared that no competing interests exist.

macromolecular complex whose subunits are conserved within alveolates but not in other lineages, that is essential for regulated exocytosis in *T. thermophila*.

## Author summary

All cells, whether single-celled protists or multicellular organisms, interact dynamically with their environments. One important mode of interaction is the release of molecules, a phenomenon called secretion, which can then modify the environment to promote the organism's well-being. Moreover, many cells have the capacity to rapidly adjust the pathways that underlie secretion, allowing them to tailor their secretory behavior in response to changes in their surroundings. A dramatic example of this is the capacity to synthesize and then store reservoirs of secretory molecules, whose eventual release is triggered when the cell senses specific environmental conditions. This phenomenon is called 'regulated exocytosis' and has been long studied in animals, because it serves as the basis for communication between different cells and tissues. Many single-celled organisms can also secrete via regulated exocytosis, and understanding the mechanisms involved could have practical consequences for developing therapies against several devastating human parasites. In this paper, we took a genetic approach to identifying factors involved in exocytosis in a single-celled protist, the ciliate *Tetrahymena thermophila*. We find that a novel gene, which appears to be present only in Tetrahymena and its relatively close evolutionary relatives, plays an important role in the pathway. Our results add another layer to recent findings that cells like Tetrahymena evolved unique mechanisms for regulated exocytosis, expanding our appreciation of cellular biodiversity.

## Introduction

Lysosomes are ubiquitous eukaryotic degradative organelles, whose membranes and contents are derived largely but not exclusively by trafficking from endosomes [1,2]. LROs (lysosome-related organelles) are a highly diverse group of cell-type specific organelles that share biosynthetic mechanisms with lysosomes [3–5]. An attribute of many LROs is the ability to fuse with the PM and release their contents to the cell exterior via exocytosis. Examples of storage in and secretion from LROs include melanin for pigmentation and photoprotection, von Willebrand factor for platelet recruitment, lytic hydrolases for immune protection, lung surfactant, processed antigenic peptides, acrosome contents during fertilization, and components for blood clotting [5–8]. When release occurs in response to specific extracellular stimuli, such as in neuroendocrine dense-core granules, the phenomenon is called regulated exocytosis [9,10].

The compounds released from LROs in humans play key physiological roles, and for that reason mutations in genes required for LRO biosynthesis or secretion cause a range of diseases. Consequently, the mapping of such disease loci has played a major role in elucidating the multi-step pathway leading to LRO secretion [11]. LROs destined to undergo exocytic membrane fusion are first transported to and tethered at the target plasma membrane, and mechanisms underlying transport, tethering and fusion have been uncovered in a range of LROs [12]. Such mechanisms are often overlapping. For example, the MyRIP/exophilin-8 complex on secretory granules can interact directly with both the actin-based motor myosin for transport, and also with actin filaments for tethering at the cell cortex [13–16]. In immune cells, the C2-domain protein MUNC13-4 serves a tethering function for secretory granules but may

also directly regulate subsequent membrane fusion by orchestrating the bundling of SNARE proteins, which is the best understood mechanism for driving bilayer fusion [17,18].

LROs have primarily been defined and studied in animals but may be found in many eukaryotic lineages [5,19,20] including Alveolates, which are comprised of protists including the apicomplexans, ciliates, and dinoflagellates [21]. In this lineage, candidate LROs are involved in conspicuous secretory responses, many in response to extracellular stimulation. In Apicomplexan parasites such as *Toxoplasma gondii* or malaria-causing *Plasmodium falciparum*, rhoptries and micronemes are secretory organelles enabling host invasion and suppression of host defenses [22,23]. Both organelles are candidate LROs, relying on endolysosomal pathways for their formation [24]. Ciliates are largely free-living and provide key ecosystem services as microbial predators, thereby shaping phenomena as varied as rumen methanogenesis and bacterial pathogenesis, while others are consequential extracellular parasites [25–29]. In ciliates, large and often elaborate secretory vesicles are present in all lineages and are likely to play a wide range of roles including active hunting, defense from predation, and encystment [30–34]. Apicomplexan and ciliate LROs share biosynthetic mechanisms, including protein targeting based upon sortilin/Vps10-family receptors homologous to those involved in LRO biogenesis in animal cells [35–38].

Only limited data are available to date on the docking and fusion of LROs in Alveolates, but they suggest the presence of features that are not conserved in animals. In the ciliate *Paramecium tetraurelia*, genes called ND (for non-discharge) are required for the exocytic fusion of LROs called trichocysts [39–42]. These cells possess an unusual plasma membrane structure at the site where each trichocyst is tethered, a rosette of intramembranous particles [43]. The analysis of ND homologs in *T. thermophila* and in Apicomplexans, as well as plasma membrane rosettes, have recently revealed the broad conservation of exocytic mechanisms within Alveolates [44]. The mechanistic role of the rosette is as yet unknown, but ND homologs in *T. gondii* show strong physical interactions with a Ferlin-family C2-domain protein TgFER2[44]. Ferlins in animals, such as myoferlin, are likely to be directly involved in membrane fusion including for LROs, though the mechanisms are obscure [45]. An unrelated family of Apicomplexan-restricted proteins with C2-like domains, called RASPS, localize to rhoptry tips where they are required for exocytosis [46].

The ND genes were discovered via forward genetic screens for *Paramecium* mutants with defects in trichocyst exocytosis. Similar approaches were pursued in the ciliate *Tetrahymena thermophila*, with screening for mutants with defects related to homologous LROs called mucocysts [47–49]. Mucocysts are ~1 μM-long rod-shaped LROs, which store secretory proteins comprised primarily of two families [50,51]. The Grl proteins (for *Gr*anule *l*attice) form the bulk of the contents, and they assemble to form a crystalline core that expands during exocytosis to propel discharge [52]. The 2nd family is made up of Grt/Igr proteins (*Gr*anule *t*ip/ *I*nduced during *g*ranule *r*egeneration)[53]. Grt1p is concentrated at one end of the rod, near the tip that fuses with the plasma membrane during exocytosis [49]. This distribution of Grt1p demonstrates that the two mucocyst ends are not equivalent, and revealed a polarity not apparent at the ultrastructural level. In contrast, most LROs in other ciliates as well as apicomplexans possess clear ultrastructural polarity, with a tip that is specialized for docking and fusion [54–56]. The function of polarized deposition of Grt1p in mucocysts is unknown, but neither it nor a closely-related paralog is required for docking or exocytosis [57].

*Tetrahymena* mutant strain MN175 was identified by screening a pool of homozygous nitrosoguanidine-mutagenized cells for mucocyst-associated defects [48]. In MN175, mucocysts with wildtype morphology accumulated in docked positions at the plasma membrane but did not undergo efficient exocytosis in response to extracellular stimulation [58]. The defect, due to a recessive mutation, could not be bypassed by directly raising cytosolic calcium,

suggesting that it acts downstream of signal transduction. In the current manuscript, we identify the causative genetic lesion in that mutant. It lies in a novel gene (TTHERM_00658810) we have named *MDL1* (*M*ucocyst *D*ischarge with a *L*amG domain). Our analysis indicates that a novel protein complex including Mdl1p, composed of subunits conserved among Alveolates, plays an essential role in mucocyst docking and exocytosis.

## Results

### Exocytic fusion of mucocysts, but not docking, is impaired in the MN175 mutant strain

Mucocysts are elongated electron-dense secretory organelles, whose contents are organized as electron-dense protein crystals (Fig 1A). They are synthesized in the cytoplasm and subsequently transported to docking sites at the plasma membrane. These sites are not randomly distributed over the cell surface but are located at fixed intervals along rib-like cytoskeletal structures called meridians (Fig 1B). For that reason, the pattern of docked mucocysts in wild-type cells appears as a loosely-ordered array when they are visualized by immunofluorescence with antibodies against either Grl or Grt proteins (Fig 1C, top row).

The pattern of mucocysts in the MN175 mutant was indistinguishable by light microscopy from that in wildtype, suggesting that the mutant has no defect in mucocyst formation or docking (Fig 1C, bottom row). The mucocysts in MN175, however, show impaired secretion in response to cell stimulation. When wildtype cells are briefly exposed to dibucaine, a calcium ionophore, virtually all mucocysts undergo exocytic fusion and release their luminal contents to the medium. The extent of release in a stimulated cell culture can be estimated by the pellet volume following centrifugation, since the released mucocyst contents form sedimentable aggregates (Fig 1D). By this measure, MN175 mutant cells showed at least a 2-fold reduction in release compared to WT. This deficiency could not be explained by a decrease in the size of the docked mucocyst pool in the mutant cells, which instead appear to accumulate slightly more mucocysts than wildtype as judged by Western blots of whole cell lysates for a mucocyst content protein, Granule lattice 1 (Grl1p) (Fig 1E). Grl1p is synthesized as the pro-protein pro-Grl1p, which is proteolytically processed to Grl1p during mucocyst maturation [59]. Wildtype and mutant cells show similar ratios of pro-Grl1p/Grl1p, consistent with the previous deduction that mucocyst maturation is not perturbed in this mutant [58].

### The MN175 mutant bears a mis-sense mutation in a novel, Alveolate-restricted gene

MN175 was directly derived from a nitrosoguanidine-mutagenized progenitor, and doubtless bears a large number of mutations that are unrelated to the mucocyst secretion defect. To reduce this mutagenic burden and increase fertility, we used conventional outcross followed by backcross to obtain UC300, a strain whose mucocyst secretion phenotype recapitulated that of MN175. We then identified the likely underlying genetic lesion in MN175/UC300 using the strategy shown in S1A Fig, consisting of an outcross followed by a modified backcross, and subsequent sequencing of pools of F2 segregants with wildtype or mutant phenotypes. The results, followed by allelic composition contrast analysis [60], allowed us to map the MN175 mutation to a ~6 Mb interval on micronuclear chromosome 4 (chr4: 3–9 Mb) (Fig 2A). This region harbored exactly two variants with predicted effects at the protein level that showed enrichment in the mutant compared to the WT pool. One of these, a G->A transition at chr4: 5,364,873, was supported by 110 out of 111 sequenced reads covering the site in the mutant pool, while the WT pool had 30 variant-supporting reads, but also 79 reads supporting the

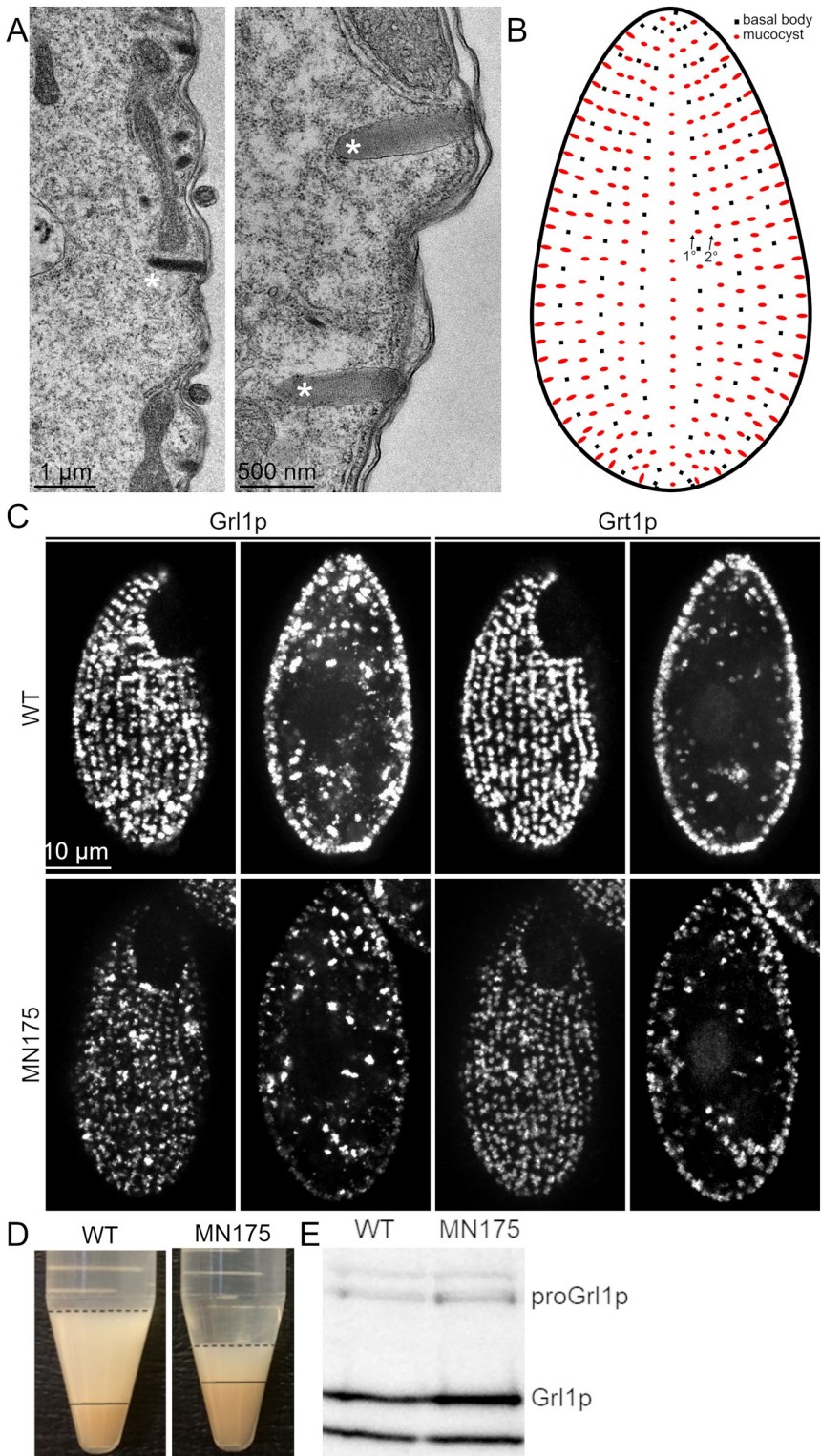

**Fig 1. Mucocyst accumulation, docking and secretion.** A. Thin section electron micrographs of mucocysts in WT cells. The large majority of mucocysts (marked with \*) are docked at the plasma membrane. The electron-dense contents are organized as a protein crystal, more easily seen at high magnification (right panel). Scale bars are indicated. B. Cartoon of a *Tetrahymena* cell, highlighting that the docked mucocysts (red) are aligned along cytoskeletal 'ribs' called 1˚and 2˚meridians. On the former but not the latter, mucocysts are interspersed with cilia,

whose basal bodies (black) are shown. C. Mucocyst visualization in wildtype cells and MN175 cells. Wildtype cells and MN175 cells were starved for 3 hrs, fixed, permeabilized and immunolabeled with antibodies against mucocyst core proteins Grt1p (which localizes to one pole of the mucocyst core) and Grl1p (which localizes throughout the mucocyst core) (4D11 and anti-Grl1p, respectively). Surface and cross-sectional images were captured with Marianas Yokogawa type spinning disk inverted confocal microscope, 100X). Wildtype and MN175 cells were indistinguishable, both showing mucocysts with the expected distribution of Grl1p and Grt1p, docked at regular intervals along meridians. Scale bar is shown. Mucocyst docking was quantitatively analyzed by calculating the fraction of total Grl1p signal intensity in cross-sectional images that is present at the cell periphery, as described in Materials and Methods. Wildtype vs. MN175 cells showed no quantitative difference in mucocyst docking: wildtype = 23%; MN175 = 23% (n = 15, no statistically significant difference by Anova: Single Factor). D. MN175 cells are defective in induced mucocyst secretion. Mucocyst exocytosis was triggered in wildtype and MN175 cells by exposing them briefly to the calcium ionophore, dibucaine. Following centrifugation, the mucocyst contents released from the cells were quantified based on the volume of the flocculent layer (dashed line) above the packed cell pellet (solid line). The relatively small flocculent layer produced by MN175 cells indicates a defect in mucocyst secretion. The wildtype cell pellet is smaller, because a larger fraction of the wildtype cells remain trapped in the flocculent layer. E. MN175 accumulates higher levels of mucocyst protein. Resolved whole cell lysates of wildtype and MN175 cells ($10^4$ cells/4-20% SDS-PAGE lane) were immunoblotted with anti-Grl1p. Compared to wildtype cells, MN175 shows approximately 35% increased levels of mature Grl1p as well as approximately 25% increased levels of the unprocessed precursor, proGrl1p. The bottom band in each lane represents a cross-reactive species that shows no change from wildtype to MN175, and provides a loading control.

reference allele (Fig 2B). The mutation is predicted to cause a change of serine residue 147 to phenylalanine in the protein encoded by TTHERM_00658810. Our interest in this candidate gene was increased by the observation that its transcriptional profile is very similar to that of genes known to be associated with mucocyst formation [57,61–65] (S1B Fig).

TTHERM_00658810 encodes a novel protein of 346 amino acids, for which we could identify homologs in other ciliates and more broadly in Alveolates, but not in fungi, animals, or plants. The protein (and its homologs) begins with a predicted N-terminal signal sequence, suggesting that it is translocated through the endoplasmic reticulum membrane (Fig 2C). Consistent with this idea, other features include a domain (Laminin G-like/Concanavalin A-like/glucanase-like) that is likely to be extracellular, based on the localization of homologous domains in characterized proteins [66], as well as the presence of the putative glycosylation site (Fig 2C). We have named the gene *MDL1* (**M**ucocyst **D**ischarge, with a **L**amG domain). However, the Mdl1p protein also contains motifs predicted to undergo modification in the cytoplasm. In particular, a near C-terminal dicysteine (C340/C341) is a candidate for palmitoylation or prenylation. The combination of features, all of which are conserved among its homologs, suggest that Mdl1p is a transmembrane protein. Consistent with this, Mdl1p residues 287–302 are predicted by MEMSAT-svm (https://bio.tools/memsat-svm) to form a transmembrane/pore-lining helix. A helical projection including that interval is shown in S2 Fig.

The mutation in *MDL1* in the MN175 strain results in substitution of serine by phenylalanine at residue 147, which lies within the predicted LamG domain (residues 110–249). Serine or threonine is conserved at that relative position in all *MDL1* homologs except those in the genus Plasmodium where the residue is alanine (Fig 2D).

## Mdl1p localizes to mucocyst tips

We added the gene encoding monomeric GFP (Green Fluorescent Protein), codon-optimized for *Tetrahymena*, to the *MDL1* open reading frame. To avoid potentially disrupting the functions of either the N-terminal signal sequence or the putative C-terminal prenylation sites, we inserted GFP after the codon for amino acid 18 in *MDL1*, following the predicted signal sequence cleavage site. This tagged allele was integrated by gene replacement at the endogenous Macronuclear locus and then driven to fixation, so that it became the sole copy of *MDL1* in the cells. Western blotting showed a single major band whose inferred molecular mass (65.0

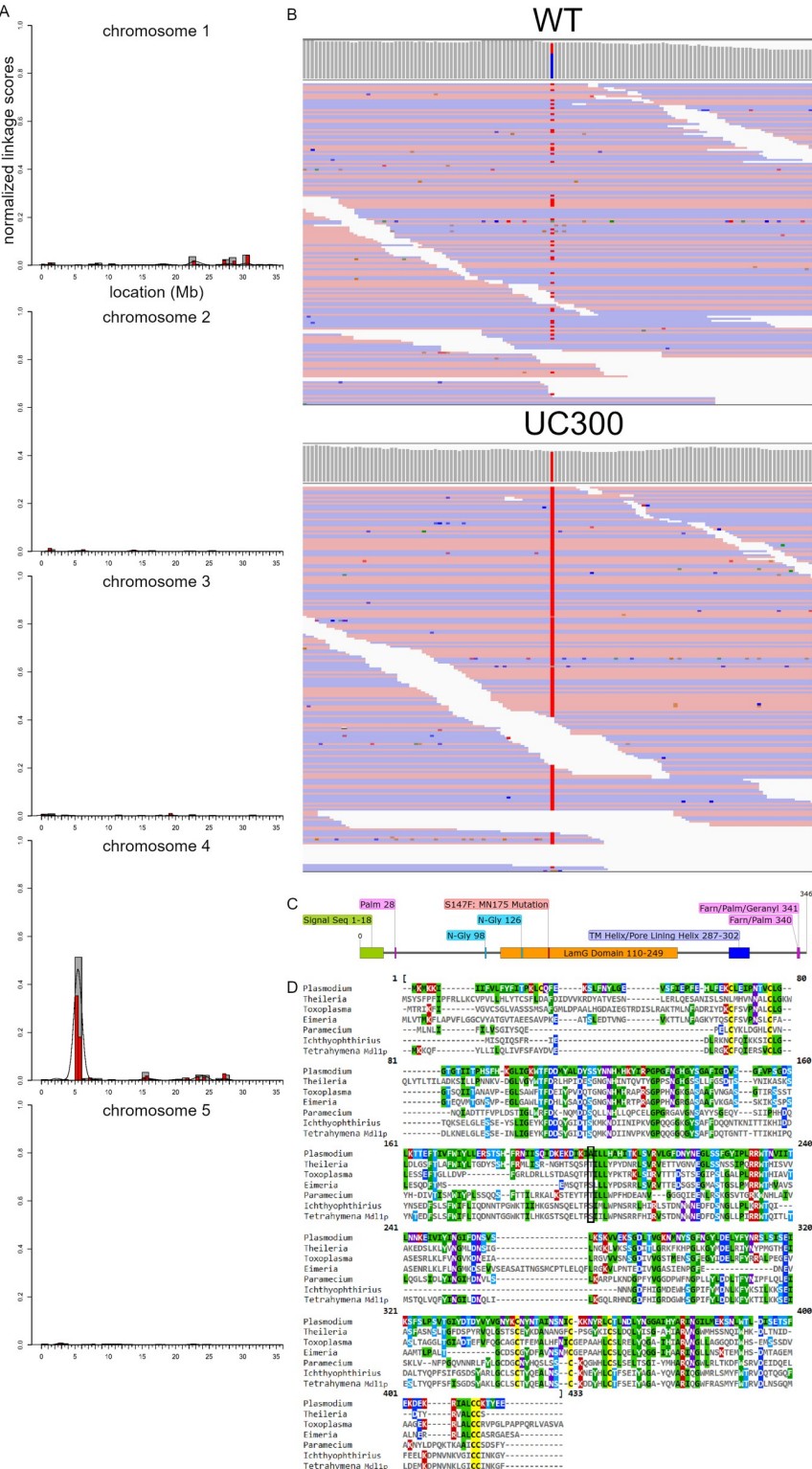

**Fig 2. Mdl1p structure and expression.** A. The results of mapping of the causal DNA sequence variant for MN175 using ACCA [60]. Normalized linkage scores are shown along all five micronuclear chromosomes. Allelic cosegregation is increased on Chr 4 3-9Mb. B. Comparison (generated with IGV [http://software.broadinstitute.org/software/igv/]) between F2 pools with WT or MN175 phenotypes, of aligned sequencing reads around chr4: 5,364,873 (highlighted). At this site, reference read is G (blue) and variant read is A (red). C. Predicted features of Mdl1p.

Residues 1–18: N-terminal signal sequence (from MEMSAT-svm). Residues 110–249: Laminin G domain (from NCBI BLASTp). Residues 287–302: transmembrane/pore-lining helix (from MEMSAT-svm). The protein lipid modification predictor GPS-Lipid (under medium threshold) identified three putative Mdl1p lipidation sites: palmitoylation at residue 28, farnesylation/palmitoylation at residue 340, and farnesylation/geranylgeranylation/palmitoylation at residue 341. The N-glycosylation predictor (NetNglyc 1.0 Server) identified two potential N-glycosylation sites at residues 98 and 126. The mutation in *MDL1* in the MN175 strain results in substitution of serine by phenylalanine at residue 147. D. MUSCLE alignment of Mdl1p with homologs (identified via reciprocal BLAST searches) from Apicomplexans (*Plasmodium falciparum* XP_001349133.1, *Theileria equi* XP_004830018.1, *Toxoplasma gondii* XP_002370635.1, *Eimeria tenella* XP_013232975.1), and Ciliates (*Paramecium tetraurelia* XP_001433723.1 and *Ichthyophthirius multifiliis* XP_004039646.1). The alignment reveals conserved features including 3 pairs of cysteines. Two pairs are located N-terminal to the predicted transmembrane/pore-lining helix, while the 3rd pair is located near the C-terminus. The residue that is substituted by the mutation in MN175 is boxed.

kDa, based on two gels) is close to the predicted 66 kDa for Mdl1p+GFP, assuming that the N-terminal signal sequence is cleaved during translocation (Fig 3A). The GFP-tagged protein appears to provide wildtype function, since strains in which it was the sole copy secreted their mucocyst contents in response to stimulation (Fig 3B).

Mucocysts in wildtype *Tetrahymena* dock along regularly-spaced cytoskeletal "ribs" that extend from the cell anterior to posterior, called meridians [67]. Mucocyst docking sites are more widely spaced on 1˚ meridians and more closely on the alternating 2˚ meridians (Fig 1B). Cells expressing GFP-Mdl1p showed an array of small green puncta at the cell surface (Fig 3C). The pattern of puncta was that expected for docked mucocysts, including differential spacing at alternating meridians. However, the green puncta were smaller as well as differently shaped from 1 μM-long mucocysts, suggesting that GFP-Mdl1p localization might be restricted to a mucocyst subdomain. To pursue this possibility we immunostained cells expressing GFP-Mdl1p with antibodies against Grl or Grt proteins. Mdl1p showed strong co-localization with a mucocyst marker (S3A Fig). Moreover, we found that GFP-Md1p appears enriched at the ends of mucocysts that are adjacent to the plasma membrane (Fig 3D). The preferred localization at mucocyst tips was supported by determining the relative alignment for many mucocysts of the GFP-Mdl1p speckle along the long axis of the Grl3p speckle (Fig 3E–3G).

Judging by its localization, Mdl1p could reside either in the mucocyst membrane or in a mucocyst-adjacent domain on the plasma membrane. To distinguish between these possibilities, we expressed GFP-Mdl1p via gene replacement in the MN173 mutant cell line. In MN173, docking-incompetent mucocysts accumulate in the cytoplasm [48]. Mdl1p in MN173 showed strong co-localization with a mucocyst marker (S3B Fig). Imaging of GFP-Mdl1p is less straightforward in MN173, where the undocked mucocysts are randomly oriented, compared to in wildtype cells. Nonetheless, by focusing on mucocysts in favorable orientations, we detected GFP-Mdl1p localized to their tips (Fig 3H). Thus, Mdl1p is a *bona fide* mucocyst protein. From this analysis, we also conclude that its polarized localization within mucocysts does not depend on prior docking.

## Mdl1p is required for mucocyst docking as well as fusion

To further explore the role of *MDL1*, we disrupted all expressed copies of the gene in the Macronucleus and confirmed the absence of the corresponding transcript in these Δ*mdl1* cells (Fig 4A). When the mutant cells were exposed to dibucaine, they failed to release any detectible mucocyst contents (Fig 4B). Thus, the gene knockout produces a more severe defect than the initial mutation in the MN175 strain. One factor accounting for this difference was revealed by microscopy. Strikingly, mucocysts in the Δ*mdl1* cells are chiefly distributed in the cytoplasm, indicating a defect in docking (Fig 4C). The mucocysts themselves have no evident defects,

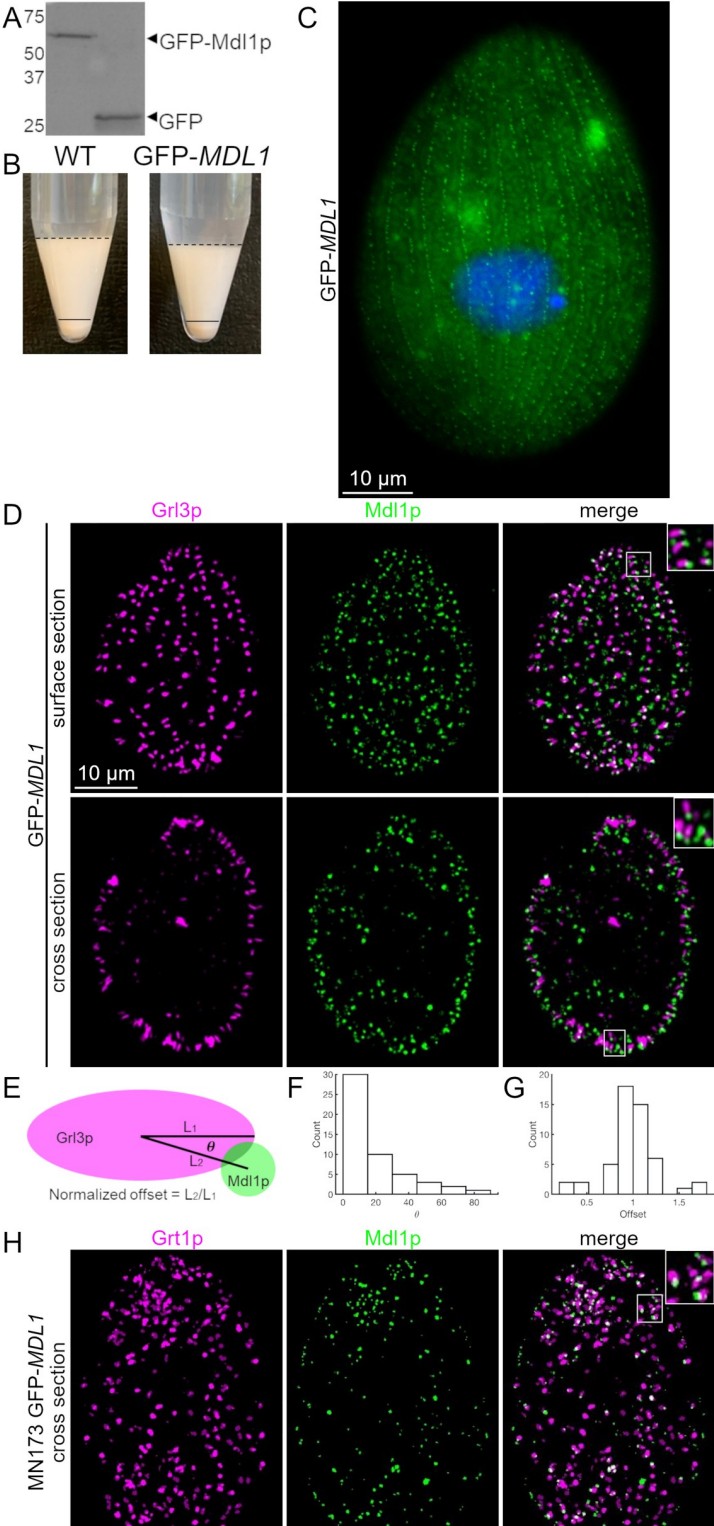

**Fig 3. Mdl1p localizes to mucocyst tips.** A. GFP-Mdl1p was immunoprecipitated from cryomilled extracts of cells expressing the transgene at the *MDL1* locus. Immunoprecipitates were resolved by SDS-PAGE and immunoblotted with anti-GFP. As shown in the left lane, GFP-Mdl1p appears as a single band of MW ~65 kDa (based on 2 gels). The positions of MW standards are shown on the left, and GFP by itself is shown in the right lane. B. Dibucaine stimulation of mucocyst exocytosis. Cells in which all copies of *MDL1* are GFP-tagged show wildtype levels of mucocyst secretion

upon stimulation with dibucaine, indicating that tagged Mdl1p retains activity. C. Live cell imaging of GFP-Mdl1p. Cells expressing GFP-Mdl1p at the endogenous locus were grown to stationary phase, fixed, and the nuclei stained with DAPI (blue). As shown here for an individual cell, small green fluorescent puncta extend in a linear array aligned with the long axis of the cell with spacing expected for docked mucocysts at 1˚ and 2˚ meridians, as explained in the text. The larger and less distinct green signals within the cytoplasm are due to autofluorescence from mitochondria and food vacuoles. Live immobilized cells were imaged using a Zeiss Axio Observer 7 microscope, with objective lens 100X. Scale bar is indicated. D. Polar distribution of Mdl1p on mucocysts. Cells expressing GFP-Mdl1p were fixed and immunolabeled with DyLight 650-conjugated antibodies against the mucocyst core protein Grl3p. In these panels, the GFP signal is colored green, and the DyLight signal is colored fuchsia. GFP-Mdl1p is concentrated at the tips of mucocyst tips where they dock at the plasma membrane, most clearly seen in the merged image inserts (right-most images). Tangential and cross sections of individual cells are shown. Scale bar is indicated. E. An explanatory cartoon depicting the angles in F and distance (offset) in G. F. Test for tip localization: histogram of the angles (˚) made by intersecting the line passing through the centroids of overlapping Grl3p and GFP-Mdl1p speckles with the long axis of the Grl3p speckle. 0˚ indicates perfect alignment of the GFP-Mdl1p speckle along the long axis of the Grl3p speckle, while 90˚ indicates perfect alignment along the short axis. G. Histogram of the distances between the centroids of overlapping Grl3p and GPF-Mdl1p speckles, normalized to half the length of the Grl3p speckle (n = 32 speckles from 6 cells). H. Distribution of Mdl1p on non-docked mucocysts. MN173 mutant cells, in which mucocysts fail to dock, were transformed to express GFP-Mdl1p at the *MDL1* locus. Overnight cultures were fixed and immunolabeled with antibodies against the mucocyst tip protein Grt1p, which is distributed broadly at the docking end of the mucocyst. Because the mucocysts are large and present in random orientations, only a small number of tips appear in each optical section. Many mucocysts show clear polarization, with GFP-Mdl1p at one tip. Cells were imaged using Marianas Yokogawa type spinning disk inverted confocal microscope, 100X. Ilastik (pixel classification software) was used to filter out diffuse autofluorescence arising from mitochondria.

since Grt1p overlaps with Grl1p (Fig 4D) and is concentrated at one end (Fig 4C, right panel insert). Consistent with the idea that the undocked mucocysts have undergone normal maturation, electron microscopic imaging of thin sections revealed mucocysts with wildtype features including elongated shape and a crystalline core (Fig 4E). Thus *MDL1* does not appear to be required for mucocyst synthesis, maturation, and polarization, but is essential for efficient docking.

The Δ*mdl1* cells appeared to accumulate a relatively small number of cytoplasmic mucocysts, as compared to the abundance of docked mucocysts in wildtype. Consistent with this, western blots of whole cell lysates indicated that Δ*mdl1* accumulates less of the mucocyst cargo protein Grl1p, particularly in starved cell cultures (Fig 4F). Decreased mucocyst accumulation may reflect degradation of undocked mucocysts in Δ*mdl1*, as suggested by images in which mucocysts appear to cluster in multivesicular bodies (S4 Fig). The difference between the defects resulting from the *MDL1* S147F substitution, compared to the complete gene knockout, raise the possibility that Mdl1p may be required for two distinct steps, docking and then fusion.

## Mdl1p is a membrane-associated protein

At the N-terminus of Mdl1p, a predicted signal sequence implies that the protein is translocated into the endoplasmic reticulum and reaches mucocysts via the secretory pathway. Therefore, extracytoplasmic domains of Mdl1p should be accessible to enzymes in the ER-Golgi lumen that catalyze N-linked and O-linked glycosylation. Mdl1p has two predicted sites of glycosylation (Fig 2C). To ask whether Mdl1p is indeed glycosylated, we treated immuno-isolated protein with PNGase F, an enzyme that removes virtually all asparagine-linked oligosaccharides [68]. Following incubation with PNGase F, Mdl1p decreased slightly in apparent MW (Fig 5A). This result supports the idea that one or more domains of Mdl1p is exposed to the lumen of early secretory compartments.

The prediction of a transmembrane/pore-lining helix suggests that Mdl1p is an integral membrane protein. To test this idea, we made cracked cell extracts and asked whether the Mdl1p is soluble, in the presence or absence of added detergent. In these experiments, Mdl1p

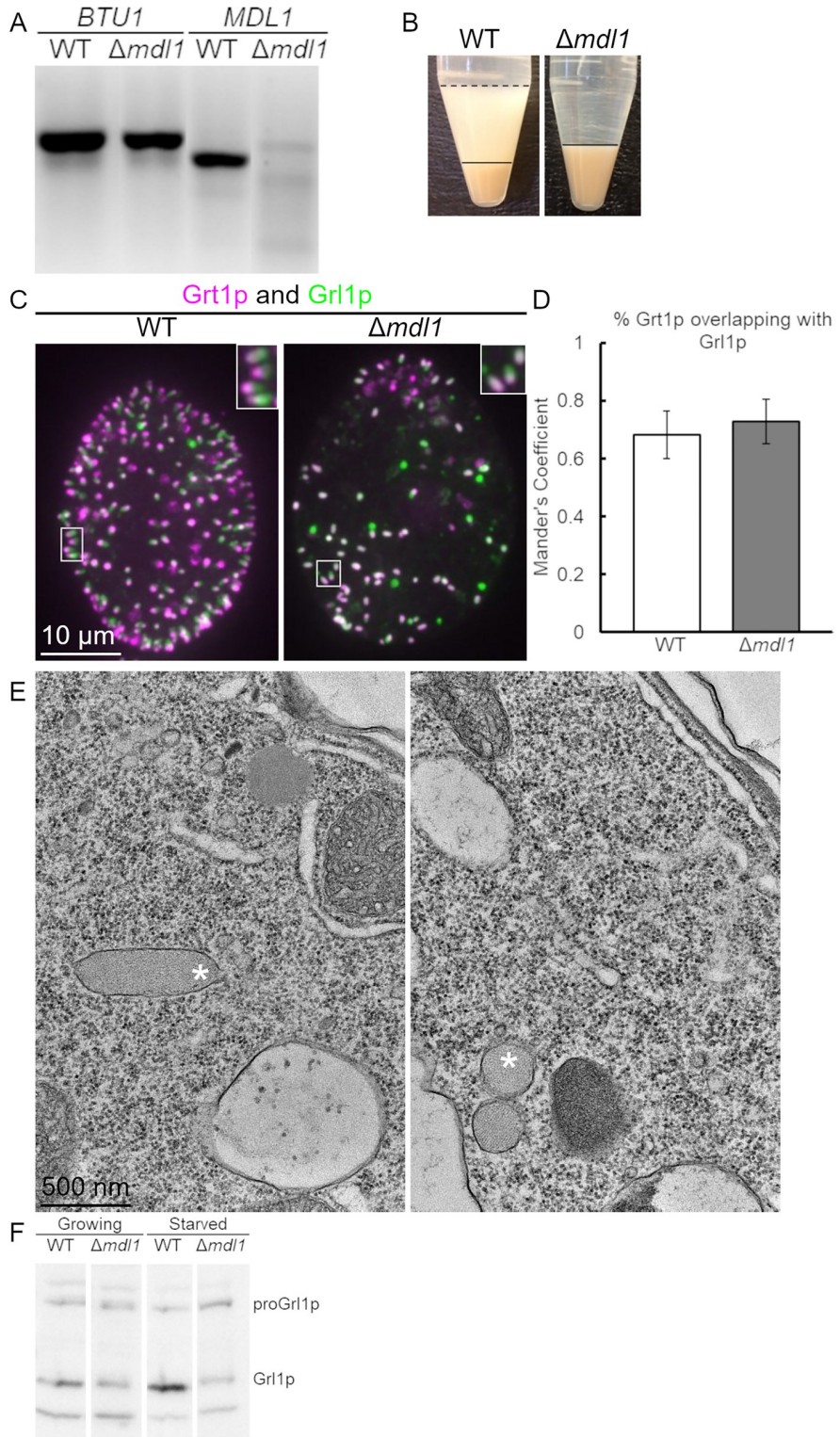

**Fig 4. Analysis of Δ*mdl1* cells.** A. *Δmdl1* cells have no detectable *MDL1* transcript. cDNAs from wildtype and *Δmdl1* were used as templates in PCR reactions with *BTU1*-specific primers as a control (lanes 1,2), and with *MDL1*-specific primers (lanes 3,4). The *MDL1*-specific primers flank intron 5. B. *Δmdl1* cells are defective in induced mucocyst secretion. Equal numbers of wildtype and *Δmdl1* cells were stimulated with dibucaine. The cell pellet is smaller in the wildtype sample compared to the mutant because in the former some cells remain trapped in the flocculent layer. The

secretion defect in *Δmdl1* is more severe than in MN175 *(mdl1-S147F)*. C. Immunolocalization of mucocysts in *Δmdl1*. WT and *Δmdl1* cells were immunolabeled with antibodies against both Grl1p (green) and Grt1p (fuchsia). Like in wildtype cells, the mucocysts in *Δmdl1* are polarized with Grt1p concentrated at one tip. In contrast with wildtype or MN175 cells, the mucocysts in *Δmdl1* are predominantly non-docked and localized throughout the cytoplasm. Scale bar is indicated. D. Quantification by Mander's coefficient of overlap between Grt1p and Grl1p in WT and *Δmdl1* cells. E. Thin section electron micrographs of mucocysts in *Δmdl1* cells. The large majority of mucocysts (marked with *) are not docked at the plasma membrane. As in WT mucocysts, the contents are organized as a protein crystal. In the right panel, the two mucocysts shown were sectioned on their short axes. Scale bar is indicated. F. *Δmdl1* contains reduced levels of mucocyst protein. Resolved whole cell lysates of wildtype and *Δmdl1* cells, from both growing and starved cultures ($10^4$ cells/4-20% SDS-PAGE lane) were transferred to PVDF, and immunoblotted with antibodies against Grl1p. Compared to wildtype cells, *Δmdl1* shows approximately 50% reduction of mature Grl1p in growing culture and 85% reduction in starved cultures.

was first concentrated by immunoprecipitation prior to running on Western blots, because otherwise its concentration was below detection limits. We found that in the absence of detergent Mdl1p was largely absent from the soluble fraction, consistent with Mdl1p being a membrane protein (Fig 5B). This was similar to but subtly different from the behavior of the type I transmembrane protein Sor4p[35], which was completely absent from the soluble fraction in the absence of detergent (Fig 5B).

Because Mdl1p has several potential sites for lipid modification (Fig 2C), such lipidation might contribute to the membrane association of Mdl1p. The most strongly predicted site for lipidation is the dicysteine at residues 340/341, so we mutated both to alanines in the GFP-tagged protein. The protein product of GFP-*mdl1*-C340,341A, expressed at the endogenous locus as the sole copy of *MDL1*, accumulated at lower levels compared to the wildtype protein (Fig 5B, compare lanes 4 and 6) and was below the detection limit for microscopy. Our inability to detect the fluorescent protein might also be explained if it were diffusely localized, rather than concentrated on mucocyst tips. The mucocysts in GFP-*mdl1*-C340,341A cells did not dock, although they displayed normal polarity with Grt1p concentrated at one end (Fig 5C; co-localization of Grl1p and Grt1p quantified in 5D). Consistent with the docking defect, the cells showed no protein release from mucocysts in response to stimulation (Fig 5E). Thus, the double cysteine-to-alanine substitutions in the GFP-tagged protein phenocopies the complete gene knockout. The residual GFP-mdl1p-C340,341A, like the wildtype protein, was only soluble in the presence of detergent (Fig 5B), indicating that these cysteines are unlikely to contribute to membrane association per se. Overall, these results suggest that Mdl1p is associated with the mucocyst membrane, but the association is less strong than for a classical transmembrane protein.

## Mdl1p is associated with a set of Alveolate-restricted proteins

To gain potential insight into the roles of Mdl1p in docking and exocytosis, we first asked whether it was monomeric or part of a larger complex. We tagged Mdl1p by integrating the FLAG peptide near the C-terminus in a stretch predicted to be disordered, N-terminal to the di-cysteine putative lipidation site. This tagging did not detectably impair protein function, since the cells in which *MDL1*-FLAG replaced endogenous *MDL1* showed wildtype levels of mucocyst secretion in response to dibucaine. We generated cryopowders from cells endogenously expressing Mdl1p-FLAG and identified favorable detergent solubilization conditions (S5 Fig) before doing pulldowns followed by elution with FLAG peptide. The eluted proteins were sedimented on glycerol gradients which were then fractionated, resolved by SDS-PAGE, and visualized by Western blotting using anti-FLAG antibodies. Molecular weight standards, sedimented in parallel gradients, provided a rough guide to estimate the sizes of Mdl1p-containing species. Mdl1p-FLAG was concentrated in two fractions (S6 Fig). The smaller fraction

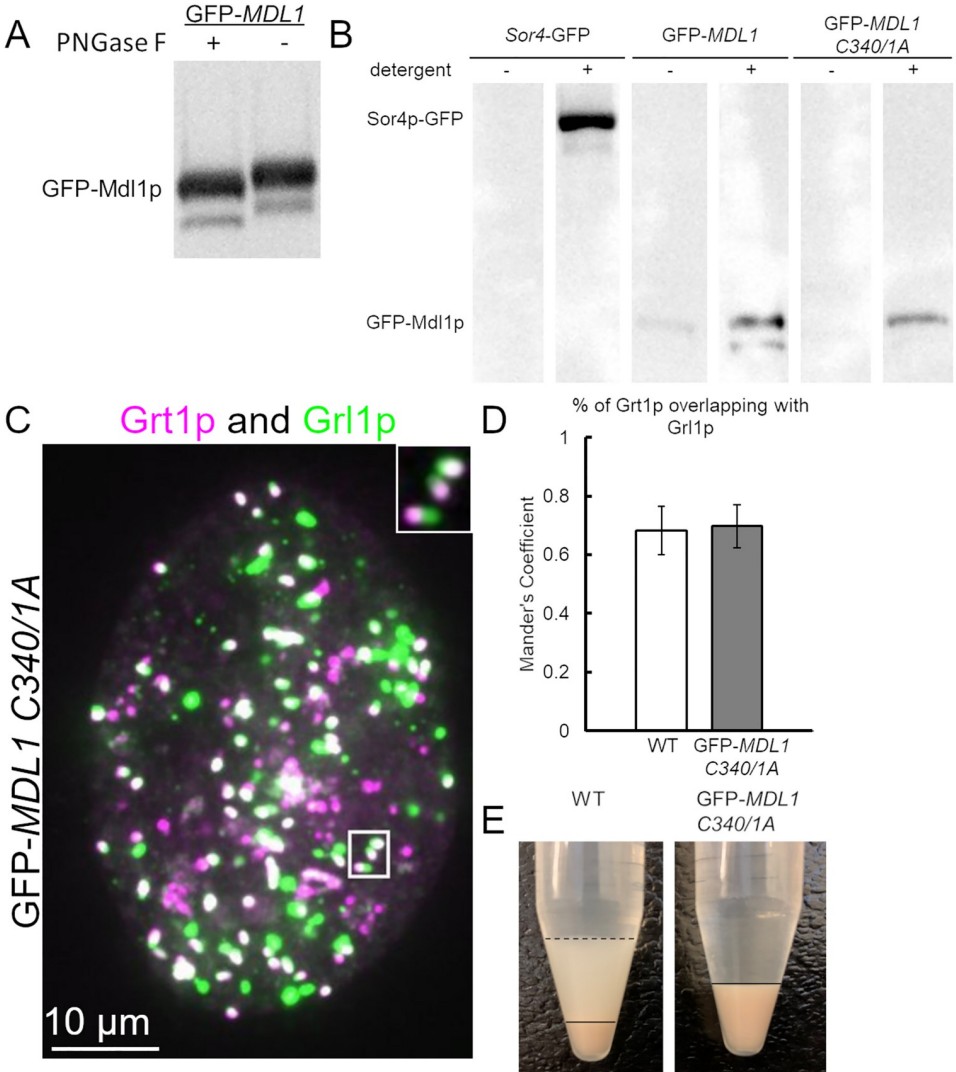

**Fig 5. Membrane association of Mdl1p.** A. GFP-Mdl1p was immunoprecipitated from detergent-solubilized cell pellets derived from 50ml cultures. Bead-bound eGFP-Mdl1p was incubated with (+) or without (-) PNGase F, eluted, resolved by SDS-PAGE, and Western-blotted with anti-GFP Ab. The increase in mobility after treatment with PNGase F indicates that Mdp1p is a glycoprotein. B. Subcellular fractionation. 50 ml cultures of wildtype cells expressing Sor4p-GFP, GFP-Mdl1p, or GFP-mdl1p C340A C341A were concentrated and lysed using a ball bearing cell cracker, as described in Materials and Methods. The lysates were treated with detergent, after which the insoluble material was pelleted by ultracentrifugation. GFP-tagged proteins in the soluble fractions were immunoprecipitated, resolved by SDS-PAGE, and immunoblotted with anti-GFP Ab. Sor4p-GFP, a Type I transmembrane protein, was found in the soluble fraction only when detergent was present. For GFP-Mdl1p and GFP-mdl1p C340A C341A, a small fraction of the protein (~6%, for the former) was soluble even in the absence of detergent, while a much larger fraction was solubilized in the presence of detergent. GFP-mdl1p C340A C341A was present at lower levels than GFP-Mdl1p. C. Imaging of mucocyst localization and polarity in GFP-*mdl1-C340A C341A* cells. Cells were starved for 3h, and mucocysts in fixed, permeabilized cells were immunolabeled with antibodies against Grl1p (green) and Grt1p (fuschia). The mucocysts are predominantly non-docked and localized through the cytoplasm. As in WT cells, the mucocysts show polarized Grt1p concentration at one tip. Scale bar is indicated. D. Overlap between Grt1p and Grl1p localization, measured by Mander's coefficient. As expected, the two mucocyst proteins show comparable overlap in wildtype and GFP-*mdl1*-C340A C341A cells. E. GFP-*mdl1* C340A C341A cells are defective in induced mucocyst secretion. WT and mutant cells were stimulated with dibucaine. The secretion defect due to the cysteine-to-alanine substitutions is equivalent to that in the complete *MDL1* knockout.

could represent a monomer or potentially dimer. The larger fraction instead corresponded to a much larger particle, whose inferred size was ~275 kDa. To identify proteins associated with Mdl1p, we repeated pulldowns of FLAG-tagged Mdl1p and analyzed total bound proteins by mass spectrometry. We found that 9 proteins were robustly associated with Mdl1p (Fig 6A) and may contribute to the large complex detected by sedimentation. Interestingly, the corresponding genes are strongly co-regulated with *MDL1*, judging by the transcriptional profiles available at tfgd.ihb.ac.cn/ (S1B Fig).

The proteins associated with Mdl1p have homologs among ciliates and sister taxa within the Alveolate superphylum, including the parasitic Apicomplexans (Fig 6B) but have no identifiable homologs in most well-studied lineages, e.g., Opisthokonta (animals, fungi) or Archaeplastida (red and green algae, land plants). All of the proteins are novel, in that no orthologous proteins have been characterized in any organism. Three of them, including Mdl1p, TTHERM_00047330, and TTHERM_000193469 have Concanavalin A/Laminin G domains. TTHERM_00141040 includes a carbonic anhydrase domain in which conservation of key residues is consistent with catalytic activity. A weak homolog of TTHERM_00141040 is involved in LRO (rhoptry) maturation in the Apicomplexan *Toxoplasma gondii*, but this protein is not orthologous to TTHERM_00141040 judging by reciprocal BLAST analysis [69].

Notably, all nine proteins, like Mdl1p, are predicted by MEMSAT-svm to possess one or more transmembrane/pore-lining domains, which are denoted and shown as helical projections in S2 Fig. We note that these are not classical hydrophobic transmembrane helices since all are predicted to include charged faces, which in some cases are extensive. To begin testing whether they constitute subunits of a complex, we asked whether three of the other putative subunits localize similarly to Mdl1p. In each case, we integrated mNeon at the C-terminus of the endogenous copy of the gene. In preliminary trials we found that tagged proteins formed punctate patterns strikingly similar to those of GFP-Mdl1p in live-cell imaging, but also that the puncta were lost upon fixation required for co-localization with mucocyst contents visualized by immunofluorescence. We therefore took an alternative approach, in both wildtype and MN173 backgrounds, of co-expressing the mNeon-tagged putative complex subunits with mCherry-tagged Igr1p, a mucocyst content protein. The co-localization analysis is less clear than for Mdl1p, particularly in the MN173 cells, because the cytoplasmic structures are in motion during live imaging and because the expression of tagged Igr1p appears to induce aberrantly large mucocyst-related structures. Nonetheless, all of the proteins were found to be unambiguously localized to docked mucocysts in wildtype cells, and present in cytoplasmic puncta overlapping with Igr1p in MN173 cells (Fig 7). These results are consistent with the idea that the proteins collectively form a complex on mucocysts, which we have named MDD for **M**ucocyst **D**ocking and **D**ischarge.

We pulled down the MDD complex using detergent-solubilized cryopowders from cells endogenously expressing FLAG-tagged Mdl1p, and eluted the bound material using FLAG peptide, as for the glycerol gradients. Eluates were applied to carbon-coated grids and negatively stained for visualization by electron microscopy (Fig 6C left panel). Particles appeared in different sizes (red and yellow circles), which could be due to different projections or sample heterogeneity, e.g., smaller particles may reflect dissociation during sample preparation. We acquired a dataset of 40 micrographs and did 2D classification of 5,164 particles using RELION [70]. The class averages showed potential top and side views of the putative MDD complex (Fig 6C right column). From the potential side view (Fig 6D), the dimensions of the particle were ~170 x 110Å. The particles were seemingly formed of two lobes with an internal channel, that may span a lipid bilayer. Interestingly the apparent openings in the lobes were asymmetric, 80Å on one side of the membrane and 50Å on the other side. These images

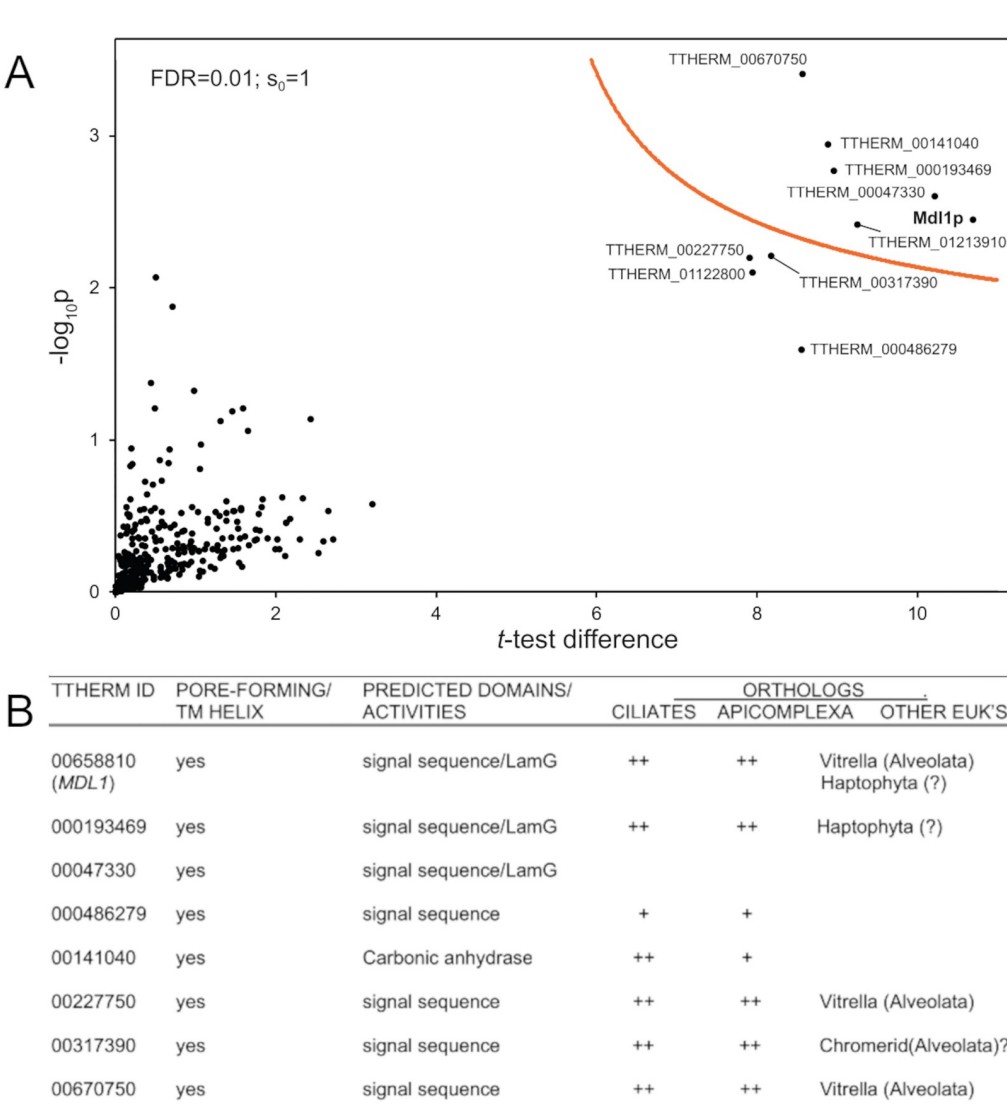

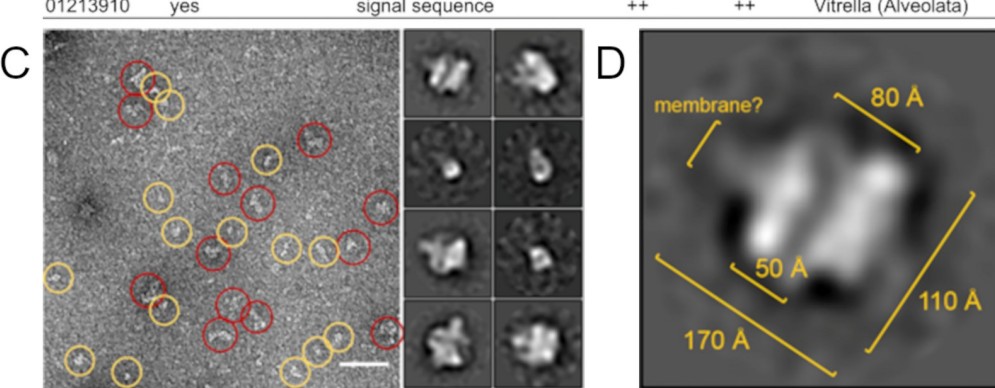

**Fig 6. Identification and analysis of Mdl1p-interacting proteins.** A. Volcano plot of mass spectrometry results, identifying the proteins associated with FLAG-tagged Mdl1p. Cryopowders from WT and Mdl1p-FLAG-expressing cells were solubilized and complexes immuno-isolated with anti-FLAG beads. Bound proteins were eluted with LDS sample buffer and run ~1cm into an SDS-PAGE gel. Individual gel lanes were excised and processed for mass spectrometric analysis. On Volcano plots such as the one shown here, proteins falling above the threshold line are considered

significant. Each sample was prepared in duplicate. B. Features of the proteins co-isolated with Mdl1p. C. Putative MDD complexes were immunoisolated as in (A) but eluted with an excess of FLAG peptide, allowed to adhere to carbon-coated grids, and then negatively stained with uranyl formate. The two rough classes of particles visualized are shown in the left panel (yellow vs red circles). Class averages acquired by 2D classification of 5,164 particles are shown in the two columns at the right, and may represent different orientations of the MDD complex. Scale bar is 50 nM. D. A potential side view derived from (C), in which the particle appears to be comprised of two lobes with an internal channel.

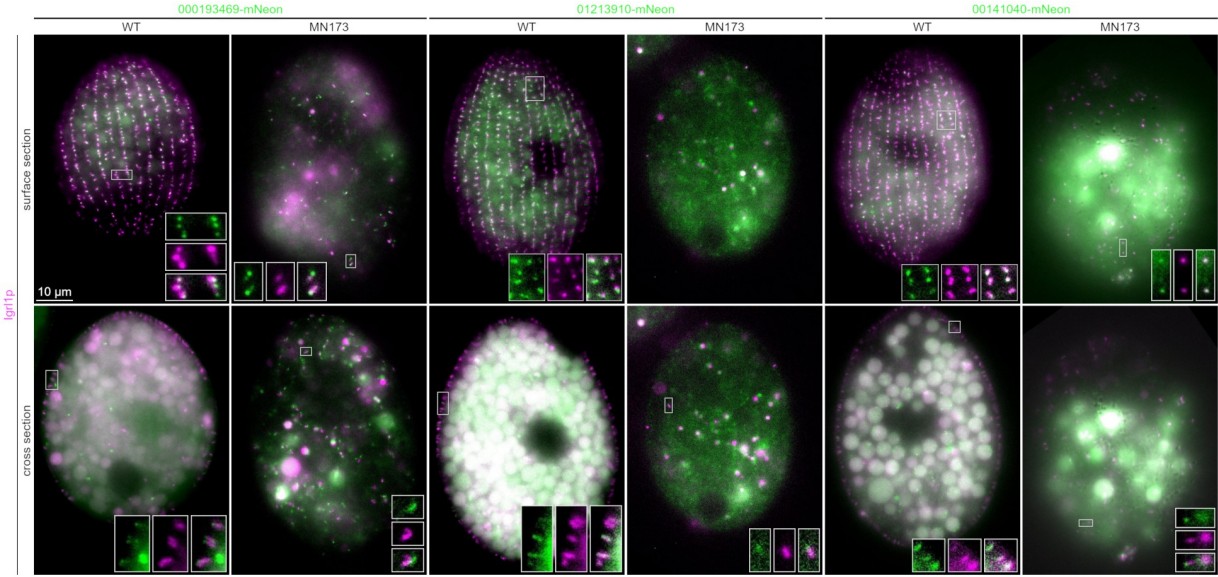

**Fig 7. Colocalization of MDD subunits and mucocysts.** Surface and cross-sectional images of live wildtype or MN173 cells expressing putative MDD subunits endogenously tagged with mNeon (green) at their C-terminus. A mucocyst content protein, Igr1p, was endogenously tagged with mCherry (magenta). Inserts show enlarged and adjusted separate and merged channels for the boxed areas. Scale bar is indicated.

reinforce our interactome data as well as glycerol gradient results in supporting a defined multi-subunit MDD complex.

## Discussion

Fusion between membrane compartments is a fundamental element of eukaryotic cellular organization and function. For the exocytosis of membrane-bound organelles involved in secretion, the process can be considered as two steps, tethering and fusion *per se* [71]. In work reported in this manuscript, we report a novel protein and complex involved in exocytosis of *T. thermophila* LROs called mucocysts. *MDL1* was uncovered via analysis of a mutant that accumulated docked mucocysts indistinguishable from those in wildtype cells, but which did not undergo exocytosis in response to cell stimulation. The gross defect in the MN175 mutant was therefore equivalent to those in several ND mutants isolated in the ciliate *Paramecium tet-raurelia* [56]. For a set of the ND mutants, analysis of the underlying genes revealed both membrane and cytosolic proteins required to establish links between the mucocyst homologs (called trichocysts in *Paramecium*) and the plasma membrane [39–42]. The list of ND genes has more recently been expanded by focusing on genes shared between ciliates and the sister lineage of apicomplexan parasites, and also exploiting isolation of physical interactors [44]. While neither the organization nor mechanism of action of the ND genes are yet elucidated, some appear to be required to form characteristic circular arrays of large integral membrane

particles that assemble at exocytic sites in Alveolates [44]. These intramembranous arrays have not been detected in other lineages and the ND genes are similarly Alveolate-restricted, supporting the idea of lineage-specific secretory adaptations.

Mdl1p, and the proteins associated with it in the MDD complex, are distinct from any of the ND genes and moreover unlikely to interact stably with them, as no ND gene products were detected as even minor species by mass spectrometric analysis of Mdl1p pulldowns. A single Alveolate homolog of an MDD subunit has previously been investigated, a non-orthologous homolog of TTHERM_00141040, and it is required for maturation of the LROs (rhoptries) in *Toxoplasma gondii* [69]. Although the precise roles of Mdl1p and the MDD complex remain to be elucidated, our data suggest strongly that the complex functions late in the pathway of mucocyst exocytosis, at the stage of mucocyst docking and fusion. In *Tetrahymena*, newly synthesized mucocysts undergo biochemical and morphological maturation [59,72]. The mucocysts in *Δmdl1* cells manifest no discernible defects in maturation, but do not stably dock at the plasma membrane and instead accumulate in the cytoplasm. The level of accumulation is lower than for docked wildtype mucocysts, and *Δmdl1* cells frequently contained what appeared by fluorescence microscopy to be clumped mucocysts within likely degradative bodies. These observations suggest that the undocked mucocysts in *Δmdl1* cells may be targeted for turnover. A different defect was present in cells expressing Mdl1p with a S147F substitution within the LamG domain. LamG domains, which fold as a β-sandwich, are often found in proteins that act as receptors, for example mediating interactions involving the extracellular matrix or between neighboring cells [73–75]. LamG-domains in the extracellular matrix of animals bind to specific glycans [75], and other characterized LamG domains bind diverse anionic carbohydrate ligands such as heparin sulfate [76]. In *Tetrahymena* Mdl1p, the LamG domain is predicted to face the mucocyst lumen, where it could potentially interact with the secretory cargo and thereby contribute to some aspect of mucocyst protein organization. Our data do not rule out a subtle contribution of *MDL1* to establishing polarized features of mucocysts that may be required for efficient exocytosis. Based on the limited characterization to date [77,78], the proteins in mucocysts are likely to be glycosylated, a conclusion also supported by the analysis of a heterologous protein targeted to mucocysts (T. Clark, pers. comm.) The LamG domain would subsequently become exposed extracellularly during exocytosis. Interestingly, glycan binding by LamG domains in the extracellular matrix is strictly calcium-dependent [75]. This may be relevant for Mdl1p, since modulation of calcium in the mucocyst lumen may regulate both biogenesis and exocytic release [79].

Taken together, our results suggest that Mdl1p is directly or indirectly involved in at least two steps in mucocyst secretion. One major question is whether the rest of the MDD complex assembles and localizes properly in the absence of Mdl1p, or if instead the disruption of the *MDL1* gene leads to destabilization or mis-targeting of multiple subunits. Thus, dissecting the functions of Mdl1p and other MDD subunits will depend on first understanding when and where this complex assembles on route to mucocysts as well as the basis for polarized tip localization. Our previous work established that protein trafficking to mucocysts involves both the classical secretory pathway as well as endosomes, a common feature of LROs [38].

All mucocyst-associated genes previously uncovered by forward mutagenesis or other approaches in *T. thermophila* are involved in mucocyst biogenesis and therefore act upstream of *MDL1* in the pathway of mucocyst secretion (reviewed in [51]). One generalization from these prior studies is that a majority of the biogenesis factors involved in protein trafficking belong to gene families where homologs in other lineages play similar roles, e.g., [61,63], though we also discovered ciliate-specific factors [80] as well as ciliate-specific expansions within conserved gene families [35,38]. In contrast, the major proteins stored within and secreted from *Tetrahymena* mucocysts are lineage-restricted, to the extent that only weak

homologs can generally be identified even in other ciliates [52,53]. *MDL1* and the other MDD subunits show a different pattern; they are conserved within ciliates and more broadly within Alveolates, but not in other protist lineages nor in animals or fungi. Since this is the same pattern as that found in the ND genes that are similarly involved at the level of exocytosis [44], our findings reinforce the idea that ancestral Alveolates had evolved unique mechanisms for the docking and exocytosis of lysosome-related secretory organelles. Intriguingly, extant ciliates and apicomplexans lack clear orthologs for key exocytosis-related genes in other lineages. Most strikingly, no study to date in Alveolates has identified SNARE proteins with the expected localization and activity expected for components involved in regulated exocytosis of LROs. In contrast, SNAREs have been demonstrated to function at many other steps in Alveolate membrane trafficking [81,82]. We note that while candidates have been advanced as factors conserved in LRO exocytosis between Alveolates and animals, those claims have curiously overlooked strongly inconsistent or contradictory published evidence, including concerning the step in which the candidate may be acting [83–86]. Taken together and in the context of ongoing work on NDD proteins in Alveolates, our results support the idea that remarkable mechanistic innovations in this eukaryotic lineage played a major role in shaping the spectrum of secretory responses.

## Materials and methods

### Cells and culture conditions

*T. thermophila* strains used are listed in S1 Table. Cells were grown in SPP [2% Bacto proteose peptone, 0.2% glucose, 0.1% yeast extract, 0.003% ferric EDTA]. Starved cells were in 10 mM Tris, pH 7.4. All growth media contained 250 μg/ml penicillin G, 250 μg/ml streptomycin sulfate, and 0.25 μg/ml amphotericin B fungizone. Reagents were from Sigma-Aldrich unless otherwise noted. Growing and starved cultures were maintained in a 30˚C shaking incubator. B*VII cells were grown in 4 ml of SPP in 60 mm petri dishes inside a humidified chamber with shaking. Cells grown in drop plates and 96 well-plates were maintained in a humidified chamber within a 30˚C stationary incubator. Cells were counted using a Z1 Beckman Coulter Counter.

### UC300 cross with IA267

For the initial outcross, 50ml cultures of UC300 and IA267 were grown to ~1.5 X $10^5$ cells/ml, washed twice and resuspended in 30˚C Dryl's medium (1.7mM sodium citrate, 1 mM $NaH_2PO_4$, 1 mM $Na_2HPO_4$, 1.5 mM $CaCl_2$). After 16–18 h, cell concentrations were adjusted to $10^5$ cells/ml, and 25 ml of each strain were combined in a stationary Fernbach flask. Pair formation was measured after 3 h at 30˚C, and 50ml of 2% proteose peptone + 0.003% ferric EDTA (2% PP) was added at 4h. Progeny were selected as described in [80]. One hour after addition of 2% PP, single mating pairs were transferred into drops of SPP (7 plates; 48 drops/plate). Plates were maintained for 2 days at 30˚C, and drops were then replicated into 96-well plates containing SPP + cycloheximide (CHX). Wells were scored for drug-resistant F1 progeny after 3 days, and 5 cells were isolated from each of 24 clones. These were grown up and serially replicated for 3 weeks, to allow ~80 fissions needed to reach sexual maturity. The clones were then tested for sexual maturity and for drug sensitivity as described in [80,87,88].

### F1 self-cross

A single fertile cycloheximide-sensitive F1 clone was induced to self-cross and thereby produce whole genome homozygotes, via conjugation with the Micronucleus-deficient strain B*VII in

the process called short circuit genomic exclusion (SCGE)[60]. In brief, F1 and B*VII were grown to ~1.5 X 10$^5$/ml, starved for 18 h, and then mixed in equal numbers to mate for 6.5 h before diluting in SPP and distributing into 96-well plates. After 18 h at 30˚C, SPP + CHX was added to the wells and 1 cell/well was isolated and transferred into drop plates. After 2 days, the drops were replicated into 96 well plates [60].

## Genomic DNA preparation

Two pools of the F2 progeny, each containing 32 clonal isolates, were used to prepare genomic DNA. The first pool consisted of clones that were phenotypically wildtype, while the second consisted of clones that were phenotypically exocytosis-deficient, i.e., MN175. The clones in each pool were combined and grown in 25 ml of SPP overnight. The pools were washed twice, resuspended in 60 mM Tris pH 7.5, and maintained at 22˚C for 2 days. 450,000 cells/sample were pelleted in Eppendorf tubes, and the supernatants removed to leave 50μl/tube. 700 μl of urea buffer (42% w/v urea, 0.35 M NaCl, 0.01 M Tris pH 7.4, 0.01 M EDTA, 1% SDS) was added. The samples were gently shaken, and Proteinase K was added to 0.1 mg/ml, followed by a 5 min 50˚C incubation. Then, 750 μl of phenol:chloroform:isoamyl alcohol (25:24:1) was added, the samples were mixed by inversion, and then centrifuged for 5 min at 10,000 x g. The top aqueous layer was transferred to a new tube, and similarly re-extracted first with phenol: chloroform:isoamyl alcohol (25:24:1), and then twice with chloroform:isoamyl alcohol (24:1). To the final extracts were added 0.2 μl of RNAse A (100 mg/ml) followed by incubation for 3 h at 55˚C, and then a final extraction with chloroform:isoamyl alcohol (24:1). In each of the final extracts, DNA was precipitated by adding an equal volume of isopropyl alcohol + 1/3rd volume of 5 M NaCl. Precipitated DNA samples were gently spooled with a hooked glass pipette, transferred to new tubes, washed twice with 70% ethanol, and air-dried for 10–15 min before resuspension in 25 μl of TE buffer pH 8.0 for storage at -20˚C.

## Whole genome sequencing

Processing and sequencing of genomic DNA was performed by Admera Health (S. Plainfield, NJ). Libraries were prepared using KAPA Hyper Prep and sequenced by Illumina (PE 150 cycle instrument) to generate 2 X 150 paired-end reads with 90X coverage. All sequencing data are available at https://www.ncbi.nlm.nih.gov/sra/?term=PRJNA817605.

## Allelic composition contrast analysis (ACCA) for the molecular identification of *MDL1*

The MiModD suite of tools (version 0.1.8 [https://sourceforge.net/projects/mimodd/] was used, with Bowtie 2(version 2.3.4.2 [89]) as the short reads aligner, for ACCA-based [90] identification of the causative mutation in the MN175 strain. The workflow for mapping the mutation, which was set up on the European Galaxy server (https://usegalaxy.eu [91]) proceeded as follows. First, Bowtie2 was used to align the sequenced reads of the F2 mutant and the WT pools to the macronuclear reference genome (GenBank assembly accession GCA_000189635 [92]). With the resulting mapped reads of both pools, joint multi-sample variant calling was performed using MiModD, and the variants annotated with predicted functional genomic effects using SnpEff (version 4.3t [93]) and its prebuilt *Tetrahymena thermophila* genome annotation file.

Prior to variant allele linkage analysis, the annotated variant list was converted to micronuclear genome (GenBank assembly accession GCA_000261185.1 [94]) coordinates and filtered for variant sites that showed at least 100x coverage in both sequenced pools with the rebase and vcf-filter subcommands of the MiModD package.

Finally, ACCA-based linkage analysis was performed with the MiModD map subcommand in Variant Allele Contrast mode, which allowed us to plot linkage scores contrasting the allelic composition of the mutant with that of the wild-type pool at each variant site against genomic (micronuclear) coordinates, and identified the peak region of the linkage signal by visual inspection of the linkage plots. To identify candidate causative variants in this region, we refiltered the full list of variants in the region for variants with 5x coverage in both sequenced pools, and for which the mutant, but not the WT pool appeared to be homozygous. Then we prioritized the candidate mutations by their predicted functional impact.

## Vector construction

*MDL1* **knockout.** The *MDL1* (TTHERM_00658810) open reading frame (ORF) was replaced in CU428 cells with the neo4 drug-resistance cassette [95] via homologous recombination with the linearized construct pNeo4-MDL1-KO. The construct contains the neo4 cassette flanked by 857 bp upstream of *MDL1* and the last 42 bp of *MDL1* plus 222 bp immediately downstream of *MDL1*. MDL1-KO was made by PCR-amplifying these upstream and downstream fragments and then cloning them into the Sac1/Pst1 and Xho1/Kpn1 sites respectively of the neo4 cassette, linearized using Sac1/Kpn1. All primers used for subcloning are listed in S2 Table.

**Endogenous GFP tagging of *MDL1*.** The ORF encoding eGFP [96] was integrated between the codons for residues 18 and 19 of *MDL1*, so that it would follow the predicted signal sequence. The *MDL1* ORF was replaced with eGFP-*MDL1*, followed by the *BTU1* terminator and the neo4 drug-resistance cassette, via homologous recombination with the linearized construct pNeo4mod-eGFP-MDL1. This vector was created by first flanking the *BTU1* terminator-pNeo4 cassette in pNeo4mod [38] with the 897 bp upstream of *MDL1* and the final 42 bp of the *MDL1* ORF + the first 222 bp of the 3'UTR, resulting in the vector pNeo4mod-MDL1-5'/3'. The upstream and downstream fragments were cloned into the Sac1/Not1 site and Xho1/Kpn1 site, respectively. To tag *MDL1* with eGFP, the eGFP tag was first amplified with the *MDL1* signal sequence from peGFP-neo4 [96] and inserted into the Pme1/Apa1 site of pNCVB-HA (gift of D. Sparvoli). The *MDL1* ORF, starting immediately downstream of the signal sequence, was then amplified from genomic DNA and inserted into the Kpn1/Apa1 site of the vector. From this vector, eGFP-*MDL1* was reamplified to add a Nhe1 site at the 3' end. This PCR product was digested with Pme1/Nhe1 and was then cloned into the Pme1/Spe1 site of pNeo4-MDL1-5'/3'. This construct was linearized using Sac1/Nae1 and used to transform CU428 and MN173 cells.

**Endogenous FLAG tagging of Mdl1p.** The FLAG tag was integrated upstream of the C-terminus of *MDL1*, between residues 332 and 333, in a region predicted to be disordered, and N-terminal to the near-C terminal dicysteine motif. The *MDL1* ORF was replaced with *MDL1*-FLAG, *BTU1* terminator and the neo4 drug-resistance cassette via homologous recombination with the linearized construct pNeo4mod-MDL1-FLAG. The FLAG tag was introduced 45 bp upstream of the C-terminus of *MDL1* via overlapping PCR. The first 1356 bp of *MDL1*+ FLAG and the last 45 bp of *MDL1*+ FLAG were separately amplified from genomic DNA, with the FLAG tag incorporated within the reverse and forward primers for the two PCRs, respectively. The amplicons were gel purified and then added into an overlapping PCR, at a 1:1 molar ratio, along with the forward primer used to generate the N-terminus + FLAG fragment and the reverse primer used to generate the C-terminus + FLAG fragment. This overlapping PCR resulted in the amplification of *MDL1*-FLAG. *MDL1*-FLAG was then cloned into the Pme1/Spe1 site of pNeo4mod-MDL1-5'/3'. This construct was linearized using Sac1/Kpn1 and used to transform CU428 and MN173 cells.

**Endogenous tagging of MDD subunits and *IGR1*.**   MDD subunits TTHERM_000193469, TTHERM_01213910 and TTHERM_00141040 were selected for endogenous tagging by 2mNeon2HA. The termination codon and a short downstream fragment was replaced by an in-frame coding fragment of 2mNeon2HA, followed by *BTU1* transcription terminator and a neo5 selection marker in reverse orientation. The tagging constructs were created using NEBuilder HiFi DNA Assembly (E2621S, New England Biolabs) and were used to transform starved CU428 and MN173 cells. After transformation, clones were assorted in increasing concentration of paromomycin, up to 1500 µg/ml, for five to ten serial tranfers and expansions. Clones were verified to express tagged MDD subunits gene products by live-imaging prior to a second transformation for endogenous tagging of *IGR1*.

The termination codon of *IGR1* was replaced by an in-frame coding fragment of 3xmCherry2HA, followed by *BTU1* transcription terminator and a pur4 selection marker (from the *Tetrahymena* Stock Center, PID000001). Six strains verified to express tagged MDD subunits (above) were starved and transformed by the construct for *IGR1* tagging. The transformants were assorted in 400 µg/ml paromomycin, 200 µg/ml puromycin and decreasing concentrations of $CdCl_2$ (from 2 to 0.5 µg/ml) for five to ten serial transfers prior to live-imaging examination.

**Site-specific mutagenesis (C340A, C341A) of eGFP-*MDL1*.**   To substitute the pair of cysteine codons at residues 340/341 with alanines, the TGTTGT sequence located between bps 1378 and 1383 was mutated to GCTGCT, by amplifying the entire gene from pNeo4mod-eGFP-MDL1with a reverse primer that contained the desired codons. The *MDL1* ORF was replaced with eGFP-*MDL1* + CC to AA mutation, *BTU1* terminator and the neo4 drug-resistance cassette via homologous recombination with the linearized construct pNeo4mod-eGFP-MDL1+C340A C341A. The PCR product was digested with Pme1/Nhe1 and then cloned into the Pme1/Spe1 site of pNeo4-MDL1-5'/3'. This construct was linearized using Sac1/Nae1 and was subsequently used to transform CU428.

## Biolistic Transformation

50 ml *Tetrahymena* cultures were grown to 500,000 cells/ml and starved for 18–24 h. Biolistic transformations were performed as described previously [38,59,97]. Transformants were identified after 3 days selection in paromomycin sulfate (120 µg/ml with 1 µg/ml of $CdCl_2$), and then serially passaged 5x/week for ~4 weeks in decreasing concentrations of $CdCl_2$ and increasing concentrations of paromomycin sulfate. The progress in Macronuclear allele replacement was judged based on the relative reduction in abundance of the cognate transcript.

## RT-PCR assessment of *MDL1* Knockout and *MDL1*-related cell lines

RT-PCR was done as described in [38]. In brief, total RNA from wildtype and gene knockout strains was isolated using NucleoSpin RNA (Macherey Nagel) from cultures at 1.5–3 X $10^5$ cells/ml. cDNA was synthesized using High-Capacity cDNA Reverse Transcription Kit (Applied Biosystems) and used as template in reactions to amplify the 3' 193 bp of the *MDL1* ORF + first 43 bp of the 3' UTR, using primers UC300-1RTendorepFW and UC300-1RTendorepREV. A fragment of β-tubulin 1 (*BTU1*) was amplified in parallel to provide a control.

## Mucocyst secretion assays using dibucaine and Alcian blue

Dibucaine stimulation of bulk cultures was performed as described in [38]. Alcian blue-stimulated secretion was used to distinguish wildtype vs mutant clones for sequencing.

For each F2 clone, 100μl was transferred to a 24-well plate (1 ml SPP/well) and incubated at 30°C for 16-18h. The cells were then pelleted, washed, and resuspended for 6h of starvation at 30°C. Following this, cells were re-pelleted in an Eppendorf microfuge, resuspended in 100μl, and transferred to 24-well plates. To each well, 33μl of 0.2% Alcian Blue 8GX was added forcefully, followed by 10 sec manual shaking. The volume was then brought to 1 ml with 0.25% proteose peptone 0.5mM CaCl$_2$. Plates were scored immediately afterwards by light microscopy for the appearance of blue-stained capsules resulting from mucocyst discharge. Clones with no visible capsules were judged to be homozygous for the mutation of interest.

## Cryomilling and immunoprecipitation

The approach was based on [98]. 10 L cultures of *MDL1*-FLAG and CU428 were grown to 2–5 x 10$^5$ cells/ml, washed once with 10mM Tris-HCl pH 7.4 and re-pelleted. Supernatants were rapidly aspirated to leave a dense cell slurry. The slurries were dripped from a volumetric pipette into liquid nitrogen, and the frozen beads collected and milled to powders using a Cryogenic Grinder 6875 Freezer Mill, and stored at -75°C. For each pulldown, 95-100g of cell powder was suspended in Buffer B3 (20 mM HEPES, 250 mM NaCl, 0.5% Triton X-100), supplemented with protease inhibitor cocktail tablets (Roche), gently mixed by inversion for 1h at 4°C, and then on ice until no solid matter was visible. The solutes were then centrifuged at 140k x g (Beckman Instruments type 45 Ti rotor) for 1.5h at 4°C, and the supernatants transferred to new tubes, for incubation at 4° C for 2 h with anti-FLAG beads (EZ view Red Anti-FLAG M2 affinity Gel, Sigma) which had been pre-washed with lysis buffer for 2 h at 4°C. The beads were then washed five times with 20 mM Tris-HCl pH 7.4, 1 mM EDTA, 500 mM NaCl, 0.1% NP-40, 1 mM DTT, 10% glycerol supplemented with protease inhibitor cocktail tablets (Roche) and eluted with 60 μl of 2X LDS sample buffer + 40 mM DTT. For cryo-EM, the beads were instead washed with B3 buffer once more, then eluted with 2.5x volume of 500 ng/μl 3X FLAG peptide (Sigma) in B3 buffer for 2 h at 4°C. For analysis by glycerol gradient centrifugation, 30 g of cell powder was used. Samples eluted with 3X FLAG peptide in B3 buffer were sedimented and fractionated as in [99].

## Mass spectrometry

The co-immunoprecipitation samples were processed as reported previously [99]. In brief, protein samples were loaded on a 4–20% SDS-PAGE gel, electrophoresed until they migrated ~1 cm into the gel, and briefly stained with Coomassie Blue R-250 solution (0.1% w/v Coomassie, 10% acetic acid, 50% methanol). A single 1 cm gel slice per lane was excised from the Coomassie stained gel, destained, and then subjected to tryptic digest and reductive alkylation. Liquid chromatography tandem mass spectrometry (LC-MS/MS) was performed by the Proteomic Facility at the University of Dundee. The samples were subjected to LC-MS/MS on a Ultimate3000 nano rapid separation LC system (Dionex) coupled to a LTQ Q-exactive mass spectrometer (Thermo Fisher Scientific). Mass spectra were processed using the intensity-based label-free quantification (LFQ) method of MaxQuant version 1.6.1.0 [100,101], searching the *T. thermophila* annotated protein database from ciliate.org [92,102]. Minimum peptide length was set at seven amino acids and false discovery rates (FDR) of 0.01 were calculated at the levels of peptides, proteins and modification sites based on the number of hits against the reversed sequence database. Proteins indistinguishable from the identified peptide sequence set were assigned to the same protein group. Perseus [103] was used to calculate P values applying t-test based statistics and to draw statistical plots. Proteomics data have been deposited to the ProteomeXchange Consortium via the PRIDE partner repository [104] with the dataset identifier PXD028372.

## Solubilization of *MDL1*-FLAG from cryopowders

To test a variety of solubilization conditions, 0.15 g of cryopowder from *MDL1*-FLAG-expressing cells was added to 0.34 ml of each buffer being tested (S5 Fig), on ice. After gentle pipetting to suspend the powders, the samples were centrifuged at 120k x g (Beckman Instruments 120.1 Ti rotor) for 45 min at 4˚C. The supernatants were transferred to fresh tubes, and each was combined with 5 μl of anti-FLAG beads pre-washed with the same buffer. After 2 h of mixing at 4˚C, beads were washed 5 times with cold 20 mM Tris-HCl pH 7.4, 1 mM EDTA, 500 mM NaCl, 0.1% NP-40, 1 mM DTT, 10% glycerol. Bound protein was eluted with 50 μl of LDS sample buffer + 40 mM DTT.

## Electron microscopy of the purified MDD complex

The MDD complex was immune-isolated from cryopowders generated from cells expressing Mdl1p-FLAG, and eluted with excess FLAG peptide as described above. Eluted material was adsorbed to carbon-coated grids. Grids were then negatively stained using 0.75% uranyl formate and washed, prior to imaging using a 120 KV electron microscope (ThermoFisher Spirit). From 40 micrographs 5,164 particles were picked, followed by 2D classification analysis using RELION [70].

## PNGase F assay

50 ml cultures of eGFP-*MDL1* cells were grown to ~7 x $10^5$ cells/ml were washed, pelleted, resuspended in buffer B6 (20 mM HEPES, 250 mM NaCl, 0.5% Triton X-100, 0.5% deoxy Big CHAPS) supplemented with protease inhibitor cocktail tablets (Roche), and then gently mixed for 1 h on ice. The lysates were cleared by centrifugation at 120k x g (Beckman Instruments TLA-100.4 rotor) for 1.5 h at 4˚C, and the supernatants mixed with 12.5 μl of GFP-Trap agarose beads (Chromotek) following the same protocol described above. After washing, the beads were incubated with PNGase F (New England Biolabs), using the denaturing condition protocol provided by the manufacturer. Protein was eluted with 50 μl of LDS sample buffer + 40 mM DTT, and then analyzed by SDS-PAGE on 8% Novex NuPAGE gels.

## Subcellular fractionation using cracked cell lysates

50 mls of cells expressing GFP-tagged Mdl1p or Sor4p were grown to $7.5 \times 10^5$ cells/ml, rapidly chilled, and subsequently processed at 4˚C. Cells were pelleted for 5 min $1500 \times g$, washed once in 20 mM HEPES pH 7.0, and repelleted. The pellet volume was measured, and the cells were then washed once with buffer B1 (20 mM HEPES, 250 mM NaCl) and resuspended in three pellet volumes of buffer B1 supplemented with protease inhibitor cocktail tablets (Roche). Cells were sheared by passing them through a ball-bearing cell cracker (H & Y Enterprise, Redwood City, CA) with nominal clearance of 0.0004 inches, until few if any intact cells remained. The lysate was split into two equal fractions, one of which had detergent added to achieve a final concentration of 0.5% Triton X-100 and 0.5% deoxy Big CHAPS. The lysates were kept on ice for 1h, and then cleared by centrifugation for 1.5h, 55000 rpm (Beckman Instruments TLA-100.4 rotor). The supernatants were withdrawn and mixed with 12.5 μl of GFP-Trap agarose beads (Chromotek) following the same protocol described above. Bound protein was eluted with 50 μl of LDS sample buffer + 40 mM DTT.

## TCA precipitation

Cultures were grown to 3 X $10^5$ cell/ml, and $10^5$ cells were pelleted, washed, and resuspended in 0.5ml of 10 mM Tris pH 7.4. Cells were precipitated with 50 μl of 100% trichloroacetic acid

and incubated on ice for 30 min. The samples were centrifuged for 15k x g at 4˚C for 5 min, washed once with ice cold acetone and repelleted. The pellets were dissolved in 2X NuPAGE LDS sample buffer, 40mM DTT and 1/10th of the dissolved precipitate was used for SDS-PAGE and Western blot analysis.

## Western blotting

TCA-precipitated or immunoprecipitated samples were analyzed by Western blotting as previously described [38]. Proteins were resolved with the Novex NuPAGE Gel system (8% or 4–20% Tris-Glycine gels, Invitrogen), and transferred to 0.2 μm PVDF membranes (Thermo Scientific). Blots incubated with anti-Grl1p serum were pre-blocked with 5% Carnation instant nonfat milk in TBST (25 mM Tris, 3 mM KCl, 140 mM NaCl, 0.05% Tween 20, pH 7.4), while blots incubated with anti-GFP or anti-FLAG antibodies were pre-blocked with 3% bovine serum albumin in TBST. Mouse mAb anti-GFP (BioLegend), rabbit anti-FLAG (Sigma), and rabbit anti-Grl1p (p40) serum [72], were diluted 1:5000, 1:2000 and 1:2000, respectively, in blocking solution. Proteins were visualized with either anti-rabbit IgG (whole molecule)-HPeroxidase (Sigma) or ECL Horseradish Peroxidase-linked anti-mouse (NA931) (GE Healthcare Life Sciences, Little Chalfont, UK) secondary antibody diluted 1:20000, and Super-Signal West Femto Maximum Sensitivity Substrate (Thermo Scientific).

## Immunofluorescent imaging of fixed cells

Cells were grown to 1.5–3 x $10^5$/ml and then fixed, or washed twice and starved for 3 h before fixation, for 30 min in ice-cold 4% paraformaldehyde in 50mM HEPES pH 7.0. The fixed cells were washed in HEPES, permeabilized in ice-cold 0.1% Triton X-100 in HEPES for 8 min on ice, then washed twice in HEPES. The permeabilization step was skipped for imaging of cells expressing eGFP-*MDL1*, to avoid detergent attenuation of the signal. Cells were then incubated for 30 min in blocking solution ($10^5$ cells, 1% BSA in TBS, 22˚C), followed by 1 h with primary antibodies diluted in blocking solution to a volume of 100 μl. For co-localization of Grt1p and Grl1p, these proteins were visualized respectively using mouse mAb anti-Grt1p (4D11, 49] and rabbit polyclonal anti-Grl1p antibodies diluted to 1:5 and 1:20,000 respectively. The affinity-purified anti-Grl1p antibodies [105] were the generous gift of Dr. Masaaki Tatsuka, Hiroshima, Japan. To label mucocysts in cells expressing eGFP-*MDL1*, the cells were immunostained with the 4D11 anti-Grt1p mAb, diluted as above, or 1:9 dilution of 5E9 anti-Grl3p mAb [52]. After incubation with the primary antibodies, cells were washed 3x with 0.1% BSA in TBS, and incubated for 30 min in 100μl of secondary antibodies. Texas Red-X-coupled goat anti-mouse IgG (Life Technologies) and 488-coupled donkey anti-rabbit IgG (Life Technologies) were diluted 1:100 and 1:250, respectively, in blocking solution. For some experiments, mAb 5E9 was directly conjugated to Dylight 488, and diluted 1:1. Cells that were immunostained with these antibodies were imaged after 3 washes with 0.1% BSA in TBS.

Cells were mounted with Trolox (1:1000) to inhibit bleaching and imaged at room temperature on a Marianas Yokogawa type spinning disk inverted confocal microscope, 100X oil with NA = 1.45, equipped with two photometrics Evolve back-thinned EM-CCD cameras, with Slidebook6 software (Zeiss, Intelligent Imaging Innovations, Denver, CO).

## Epifluorescent imaging of live cells

For live imaging, cells were washed and resuspended in 10 mM Tris pH 7.4 and placed directly on glass slides, where some cells become immobilized by the pressure of the cover slip. Alternatively, cold CyGel (Abcam) was buffered with 1 M Tris, pH 7.4 (100:1, for a final concentration of 10mM), and this was mixed 1:1 with cells on coverslips, which were then gently pressed

onto slides at room temperature. Images were captured with an Axiocam 702 mono using the Zeiss Axio Observer 7 system, with a 100X objective (Alpha Plan-Apo 100x/1.46 oil DIC M27).

## Quantifying mucocyst docking in wildtype and MN175 cells

For each cell, the peripheral area of a cross-sectional image stained with anti-Grl1p antibodies was selected in Fiji-ImageJ, and the signal intensity therein was divided by the overall signal intensity of the cell measured by an oval selection. The Grl1p peripheral signal intensity percentages of 15 wildtype cells were compared to 15 MN175 cells by Anova: Single Factor function in Microsoft Excel.

## Analysis of overlap between Grl-labeled mucocysts and Mdl1p signal

To measure the overlap between Grl and Mdl1p signal, we automatically segmented micrographs acquired at the surface planes of cells. Features of interest were isolated from raw images using custom MATLAB scripts to perform adaptive thresholding and morphological operations. The fraction of Mdl1p features that overlap a Grl feature was computed for sets of Mdl1p features identified using a range of sensitivity parameters for adaptive thresholding to show that more restrictive thresholding leads to greater overlap and that overlap remains high for a wide range of values before false positives dilute the signal. Random expectation for overlap was computed by randomly translocating Mdl1p features within the cell and determining whether they overlapped Grl features.

## Analysis of the localization of Mdl1p signal to the tips of Grl-labeled mucocysts

To systematically measure the degree to which Mdl1p localizes to the tips of Grl-labeled mucocysts, we automatically segmented micrographs acquired at the midsection of cells, where Grl-labeled mucocysts are elliptical and oriented perpendicular to the cell cortex at the periphery of the cell. Mdl1p features were segmented using a sensitivity parameter for adaptive thresholding that maximizes true positives, while still maintaining fractional overlap that is significantly higher than random expectation. Two quantities were used to describe the localization of Mdl1p relative to overlapping Grl-defined mucocysts: 1) the orientation of Mdl1p relative to the long axis of the mucocyst, and 2) the distance between the centroids of Mdl1p and Grl features. The first quantity was calculated by measuring the angle made by intersecting the long axes of Grl features to the line segments connecting the centroids of overlapping Grl and Mdl1p features. A value of 0 indicates perfect alignment of a Mdl1p feature along the long axis of a Grl mucocyst, while a value of 90 indicates perfect alignment with the short axis. The second value was calculated by measuring the distance between the centroids of overlapping Grl and Mdl1p features, normalized to half the length of the long axis of the Grl mucocyst. A feature that is at the tip of a mucocyst will have a value of 1. Random expectations for both values were computed by randomly translocating Mdl1p features within paired Grl mucocysts. The two-sample Kolmogorov-Smirnov test was used to assess the statistical significance of results relative to the random expectation.

## Gene expression profiles

Gene expression profiles were downloaded from the *Tetrahymena* Functional Genomics Database (TFGD, http://tfgd.ihb.ac.cn/)[64,65]. In the graphs shown, each profile was normalized by setting the gene's maximum expression level to 1.

## Supporting information

**S1 Fig. Mapping the MN175 mutation leads to *MDL1*, a gene whose transcriptional profile is similar to mucocyst-cargo genes.** (A) Generating pooled genomic DNA of WT and MN175 clones for ACCA. MN175 mutant and UC300 strain used in this study were previously described. F2 clones of UC300 were generated and tested for exocytosis (see Materials and methods). Clones with defective exocytosis were further screened for processed Grl1p similar to Fig 1E. WT-like and MN175-like clones (32 each) were pooled to generated genomic DNA used in whole genome sequencing followed by ACCA. (B) Comparison of transcriptional profiles of *MDL1* with genes encoding established mucocyst cargo proteins and proposed MDD subunits. Transcript levels are measured in growing cultures at three densities, followed by successive time points following starvation, and then during successive time points during conjugation, from tfgd.ihb.ac.cn. Mucocysts cargo protein genes include *GRL1* (TTHERM_00527180), *GRT1* (TTHERM_00221120) and *IGR1* (TTHERM_00558350). Putative MDD subunits are listed in Fig 6B. Missing from the transcriptional analysis are TTHERM_000486279 and TTHERM_000193469, for which transcriptional profiles are not publicly available.
(TIF)

**S2 Fig. Predicted transmembrane/pore-lining residues of putative MDD subunits and their helical projections.** Predicted transmembrane/pore-lining residues of putative MDD subunits and their helical projections. For each protein, MEMSAT-SVM was used to predict membrane helices with the sequence of the full protein or with the signal peptide removed. Mdl1p and all putative MDD subunits are predicted to possess one or more 16-aa intervals that are transmembrane/pore-lining. For each protein sequence, a turn of α-helix (18 aa) including the interval predicted to be transmembrane/pore-lining by MEMSAT-SVM was used to generate a helical wheel (heliquest.ipmc.cnrs.fr). Residues are colored based on polarity and the arrow indicates hydrophobic moment (μH).
(PDF)

**S3 Fig. GFP-Mdl1p partially colocalizes with Grl3p in WT and MN173 cells.** (A) The fraction of GFP-Mdl1p that overlaps with Grl3p in WT cells, at a sensitivity of adaptive thresholding appropriate to local maxima, compared to the overlap between randomized centroids within cell boundaries with Grl3p. Error bars indicate standard error of the mean (n = 6). (B) Same as A, but in MN173 cells.
(TIF)

**S4 Fig. Undocked mucocysts appear to clump in *Δmdl1*.** Undocked mucocysts appear to clump in *Δmdl1*. Starved *Δmdl1* cells were immuno-stained with mouse Abs against Grl1p and Grt1p that were directly dye-coupled, as in Fig 4C. Shown are the individual channels and the merge. Clustering of mucocyst-related puncta (arrows) suggests they are incorporated in degradative bodies. Scale bar is indicated.
(TIF)

**S5 Fig. Testing of solubilization conditions for Mdl1p-FLAG from cryopowders.** Testing of solubilization conditions for Mdl1p-FLAG from cryopowders. The solubilization assay was performed as described in Materials and Methods. Loaded in each lane is the soluble fraction of Mdl1p-FLAG, revealed by Western blotting with an anti-FLAG antibody, for the buffer composition shown in the table beneath.
(TIF)

**S6 Fig. Sedimentation analysis of the Mdl1p-containing complex.** (A) Sedimentation analysis of Mdl1p. Mdl1p-FLAG, eluted with FLAG peptide from pulldowns, was sedimented on

10–40% glycerol gradients. Fractionated gradients were analyzed by SDS-PAGE followed by Western blotting with anti-FLAG antibody. Shown are fractions 4 to 17, in order of increasing density (of a total of 50 fractions). Fractions 4–5 and 13–14 contained the peak concentrations of Mdl1p-FLAG. Shown beneath are the relative intensities of the Mdl1p signals across the gradient fractions, normalized to the maximum after background subtraction. The peak fractions for size standards run on parallel gradients are indicated with underlining: BSA (66 kDa, fraction 6); yeast alcohol dehydrogenase (150 kDa, fraction 11); thyroglobulin (660 kDa, fractions 24–26). (B) Plotting the molecular weights of the standards as a function of their corresponding peak fractions suggests fraction 13–14 correspond to a molecular weight of 260–290 kDa. (TIF)

**S1 Table.** *T. thermophila* **strains used in this study.**
(DOCX)

**S2 Table. Primers used for in this study.**
(DOCX)

**S1 Dataset. Numerical data for graphs and summary statistics.**
(XLSX)

## Acknowledgments

AK thanks Daniela Sparvoli for extensive technical help, and APT thanks Maryse Lebrun, Marta Cova, and Daniela Sparvoli for insightful comments on the manuscript. At the University of Chicago, we thank staff at the Advanced Electron Microscopy Facility for their help in EM data collection, as well as staff at the Integrated Light Microscopy Core Facility. Codon-optimized eGFP was the generous gift of Dr. Kazufumi Mochizuki (Montpellier). Affinity-purified anti-Grl1p antibodies were a generous gift of Dr. Masaaki Tatsuka, Hiroshima, Japan. MZ thanks the Fingerprints Proteomics facility at the University of Dundee for excellent support. Bioinformatics by WM was done on the European Galaxy server (https://usegalaxy.eu).

## Author Contributions

**Conceptualization:** Aaron P. Turkewitz.

**Data curation:** Wolfgang Maier, Minglei Zhao, Martin Zoltner.

**Formal analysis:** Wolfgang Maier, Charles F. Lang, Minglei Zhao, Martin Zoltner.

**Funding acquisition:** Wolfgang Maier, Mark C. Field, Martin Zoltner, Aaron P. Turkewitz.

**Investigation:** Aarthi Kuppannan, Yu-Yang Jiang, Chang Liu, Charles F. Lang, Chao-Yin Cheng, Martin Zoltner.

**Methodology:** Yu-Yang Jiang, Wolfgang Maier, Charles F. Lang, Minglei Zhao, Martin Zoltner, Aaron P. Turkewitz.

**Project administration:** Aaron P. Turkewitz.

**Resources:** Wolfgang Maier, Mark C. Field, Minglei Zhao, Martin Zoltner, Aaron P. Turkewitz.

**Software:** Wolfgang Maier, Charles F. Lang.

**Supervision:** Aaron P. Turkewitz.

**Validation:** Aarthi Kuppannan, Yu-Yang Jiang, Martin Zoltner, Aaron P. Turkewitz.

**Visualization:** Aarthi Kuppannan, Yu-Yang Jiang, Wolfgang Maier, Chang Liu, Charles F. Lang, Chao-Yin Cheng, Minglei Zhao, Martin Zoltner.

**Writing – original draft:** Wolfgang Maier, Minglei Zhao, Martin Zoltner, Aaron P. Turkewitz.

**Writing – review & editing:** Yu-Yang Jiang.

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
