## [Decision Letter · Decision Letter 0]

2 Nov 2021

Dear Dr Turkewitz,

Thank you very much for submitting your Research Article entitled 'A novel membrane complex is required for docking and regulated exocytosis of lysosome-related organelles in Tetrahymena thermophila' to PLOS Genetics.

The manuscript was fully evaluated at the editorial level and by independent peer reviewers. The reviewers appreciated the attention to an important topic and the pioneering new insights that you have presented. The manuscript is also beautifully written and will be accessible to a wide audience. You will note, however, that the reviewers and editors have identified some concerns that we ask you address in a revised manuscript.

We therefore ask you to modify the manuscript according to the review recommendations. Your revisions should address all the specific points that have been made. In providing some guidance in addressing these points , we have provided guidance in this and the following paragraphs. For point 1 of reviewer 1, if we are not mistaken, lines 188-192 in the submitted manuscript address this issue. If this is the case, please indicate so in your response letter, or if not, it will be necessary address this issue further. Point 2 of reviewer 1 is essential to address. Regarding point 3, in addition to addressing the issue of western blot quantification for Figs 1E and 5B, the editors wonder with respect to Fig 1B, whether western blotting is the appropriate approach for assessing “no change in docking” as stated in the relevant text. While the fluorescence micrographs are convincing that docking in the mutant is not changed qualitatively, please consider some quantitative measure (docked mucocysts per square micron?) here. Regarding point 4, it is essential to clarify the presentation regarding the “wild type and F2 phenotypes” for a generally interested readership. The rest of the points raised seem straightforward and addressing them is essential.

Regarding the critique provided by reviewer 2, the first point raises an unlikely complication stemming for protein interactions with the FLAG epitope tag rather than with Mdl1p. The authors already present compelling text indicating that the transcriptional profiles for Mdl1 and the interaction partners all follow the profiles that the Turkewitz lab has shown previously for mucocyst/exocytosis-related proteins. Thus, rather than employing a different epitope tag in repeating the experiments, please consider showing the transcriptional profiles (with a relevant previously characterized one as control) in a supplementary figure. This approach has the added advantage that it addresses point 5 raised by reviewer 2 concerning use of “data not shown” in the text. Point 2 raised in this review is essential to address. In doing so, the editors suggest that you use expression of the neon green-labeled protein in strain MN173 (as used earlier in the manuscript) to judge definitively that this new protein also is mucocyst (and not plasma membrane)-associated. Point 3 is also essential to address. Fig S5 in its present form is not particularly accessible to readers. Densitometry profiles across the fractions for the FLAG western (as well as controls with MW indicated in the figures) has the added advantage of documenting what appears to be a gaussian distribution for the complex, which would support its likely homogeneity. Moreover, it would further provide confidence that the negative-stained images shown in Fig 6D,E are indeed likely to show different views of the same complex. As an additional contribution to the manuscript, the editors strongly encourage inclusion of a figure panel showing a silver- (or Coomassie-) stained SDS gel profile for the complex. This would enable the authors to document purity and that the full complement of bands is consistent with an estimated molecular mass of 330 Kd. Point 4 regarding the disposition and orientation of Mdl1p will likely be settled in definitively addressing point 2 of this reviewer. Finally, regarding the “Minor point” made at the end of the critique, the editors would like you to present a helical wheel (or similarly insightful) depiction for the predicted transmembrane span of Mdl1p. This will clarify what is meant by the currently vague term “helix/pore” as stated in the Abstract and the text. All the more important, it apparently applies to the other candidates of the protein complex. We view this as a key addition that is justified by the level of information that is currently available.

Regarding the critique provided by reviewer 3, the first comment regarding Figure 2D likely stems from not examining the high-quality Tiff image provided with the manuscript. While the version included in the submitted manuscript is indeed low-resolution, the Tiff image is satisfactory. The other comments provided by this reviewer need to be addressed.

[LINK]

Yours sincerely,

J. David Castle, Ph.D.

Guest Editor

PLOS Genetics

Gregory P. Copenhaver

Editor-in-Chief

PLOS Genetics

Reviewer's Responses to Questions

**Comments to the Authors:**

Reviewer #1: This beautifully written manuscript walks the reader through (1) cloning a gene required for Tetrahymena mucocyst discharge identified by a Mendelian mutation, (2) showing that the corresponding Mdl1 protein is a transmembrane glycoprotein located at one end of the mucocyst, (3) demonstrating that Mdl1p is required for both docking and fusion steps of the mucocyst pathway and (4) using the protein to characterize a protein complex that likely constitutes a pore in the mucocyst membrane. Protein homology suggests conservation of the pore complex in ciliates and apicomplexan parasites but not in other eukaryotes. Although just a starting point for further investigation of the biogenesis, structure, function and evolution of the novel pore complex, this manuscript -- based on a wide range of genetic, molecular, biochemical and cellular techniques -- merits publication in PLoS Genetics. I will be particularly interested in learning more about the biogenesis pathway, given that mucocysts and related secretory organelles in the alveolata clade are polarized, lysosyme-related organelles (LRO) and that the pore complex is asymmetrically located at the tip of the mucocyst.

I have only minor comments (except for the last point about data availability which is essential).

1. The argument that Mdl1p is required at two steps of the pathway, docking of the mucocysts and release of the contents in response to a stimulus, is based on the MN175 mutant phenotype. In the KO line with no Mdl1p, there is no docking and a role in fusion cannot be assessed. Elsewhere in the text it is explained that the original MN175 mutant was screened from a population subjected to mutagenesis with nitrosoguanidine and therefore likely contains many other mutations. Can the authors confirm that the partial MN175 phenotype that discriminates docking from fusion is obtained with the back-crossed line (maybe I missed something)? It would be elegant to check the phenotype with a GFP-transgene bearing the same Ser -> Phe amino acid change as Mn175, after gene replacement in the WT reference strain.

2. In Figure 2, a Muscle alignment of Mdl1p with homologs is shown. Yet I could not find in the legend or in the Methods, how these homologs were identified. I am guessing some kind of blastp search at NCBI? This should be specified clearly and accession numbers of the aligned proteins should be provided, at least in the figure legend.

3. At several points in the manuscript, protein levels are compared on western blots, and it is stated that under one condition there is more protein, but there is no quantification. For example, in Figure 1E, with ‘a cross-reactive species’ -- which is vague indeed -- as loading control. Is there any way to make this (and other blots) more quantitative? So you can say there is 1.5X more protein or whatever. Similar situation in Figure 5B.

4. Figure 2B, it would be helpful to reword the legend to make it clear that ‘WT’ and ‘MN175’ F2 pools refers to phenotype e.g. F2 pools with WT and MN175 phenotypes. I also stumbled at the beginning of the methods section “Genomic DNA preparation” it must be crystal clear that ‘WT’ refers to phenotype of F2 clones and not to WT cells. “32 wildtype and 32 separate F2 clones were pooled and grown in 25 ml of SPP overnight ” can be hard to understand for someone not working with these critters every day, please reword so it is perfectly clear that there is one pool of 32 F2 clones with WT phenotype and a second pool of 32 F2 clones with MN175 phenotype..

5. Line 211, there is a missing parenthesis : “…(highlighted. At this site….”

6. line 232 missing article before 'putative' : “…as well as the presence of putative glycosylation site…”

7. line 764 “33 ul” (Greek micron character should be used as elsewhere).

8. Figure 1A an asterisk is mentioned but I could not see it in the image without blowing it way up on my computer screen. Furthermore, the legend states “the large majority of mucocysts (marked with *)” and thus it is a surprise to find only one that is marked on this image. Can the legend be reworded to convey the fact that only one mucocyst is so labeled in this image though a majority are similarly docked in the cell? Can the asterisk be made more conspicuous than black on black?

9. DATA AVAILABILITY. The Illumina sequencing data (F2 pools) generated in this study should be deposited in GenBank and an accession number included in the manuscript.

Reviewer #2: Aveolates have evolved a variety of free living, symbiotic, and parasitic lifestyles with divergent biology but common attributes including the regulated exocytosis of lysosome related organelles (LROs). Although some insight has been gained into the molecular bases of LRO biogenesis, docking, and exocytosis, much remains to be learned about these processes and events. In Tetrahymena, several mutants have been identified with defects in the biogenesis of mucocyst LROs; however, virtually nothing is known about proteins required for docking and discharge. The current study identifies from a chemical mutagenesis screen a Tetrahymena glycoprotein, Mdl1p, that is required for both docking and exocytosis of mucocysts. The investigators confirm the role of Mdl1p by making a fully null strain, they localize Mdl1p to the sites of exocytosis in polarized association with mucocysts, they identify putatively interacting proteins, and they provide evidence that Mdl1p is associated with a large protein complex that they visualize with single particle cryoEM. Importantly, several of the co-immunoprecipitating proteins have uncharacterized homologs in other aveolates. This raises the prospect of the complex being a conserved and fundamental component of docking and exocytosis of LROs in related organisms including one’s of medical, veterinary, and ecological importance. Overall, the findings are of general interest and considerable novelty, and the study advances the field in an important way.

Main comments

Although the main conclusions are generally well supported by the data, several aspects were authenticated to a satisfactorily degree.

1. The gradient centrifugation and cryoEM analyses suggest that Mdl1p is in a protein complex, possibly with multiple copies of Mdl1p. In this situation it is an unlikely but still a non-zero possibility that the co-immunoprecipitating proteins are interacting with the FLAG tag rather than with Mdl1p itself. Such a potential artifact is typically addressed by using different affinity tags to immunoprecipitate the bait protein with one or more of the prey proteins.

2. The authors tag one of the co-IPing proteins (01213910) with mNeonGreen and suggest it has a distribution that is consistent with it being associated with mucocysts, and their tips, but this remains largely speculative. The precise localization of 01213910 should be validated by IFA with known mucocysts markers.

3. The complex containing Mdl1p was estimated to be 330 kDa, but limited substance was provided to support this estimate. It would be helpful to quantify the distribution of standards and Mdl1p to more transparently so the basis of the estimate.

4. The authors suggest that the laminin domain of Mdl1p is oriented toward the lumen of the mucocyst, which indicates they are thinking that Mdl1p is embedded in the mucocyst membrane. However, it is unclear to this reviewer based on the data provided how a distinction can be made between their suggested scenario versus Mdl1p residing on the surface of the cell at sites of docking. Perhaps the authors are relying on the transcriptional profile, but the Mdl1p profile is not shown in relationship to known mucocyst or surface proteins.

5. Point 3 also gets to another aspect of concern, namely the multiple instances where data is not shown. This reviewer is not familiar with the PLOS Genetics policies on data not shown but is a proponent of showing everything that is worth mentioning.

Minor

1. Given there is no direct evidence that Mdl1p forms a helical pore, it is best to describe the transmembrane domain as such alone.

Reviewer #3: In the manuscript “A novel membrane complex is required for docking and regulated exocytosis of lysosome-related organelles in Tetrahymena thermophila” Kuppannan et al. provide insight into the mechanism of regulated exocytosis via lysosome-related secretory organelles (mucocysts) in Tetrahymena thermophila. Using a combination of forward and reverse genetics, as well as GFP tagging of endogenous MDL1 copies followed by immunolocalization experiments, the authors demonstrate that a novel Tetrahymena protein, Mdl1p, plays an essential role in docking of the mucocysts at the plasma membrane and secretion of mucocyst contents. Furthermore, Kuppannan et al. show that Mdl1p co-purifies with 9 other proteins, one of which co-localizes with Mdl1p at the tips of mucocysts, as determined by immunofluorescence. This is a solid study that lays the foundation for the future research on regulated exocytosis, and it is of high importance to scientists in the fields of protein trafficking/secretion. I recommend this manuscript for publication in PLOS Genetics; several minor points, however, need to be addressed prior to publishing.

1. The image in Figure 2D is low in quality.

2. Lines 239-240 – “the mutation in MDL1 strain results in substitution of phenylalanine for serine at residue 147” – isn’t it a substitution of serine for phenylalanine?

3. In the immunoprecipitation experiments, 140,000 x g was used to spin down the insoluble fraction (line 778). In Line 813, for the centrifuge speed value, please provide G Force value instead of rpm.

4. Grt1p is “concentrated at one end of the rod, near the tip that fuses with the plasma membrane during exocytosis” (lines 108-109), while Grl3p is localized throughout the mucocyst and Mdl1p is at the tip. However, in Figure 3H, Grt1p and Mdl1p only partially co-localize. It needs to be clarified whether Grt1p is distributed over a wider area at the end of the mucocyst rod.

**Have all data underlying the figures and results presented in the manuscript been provided?**

Reviewer #1: **No: **need to deposit sequencing data

Reviewer #2: **No: **

Reviewer #3: Yes

PLOS authors have the option to publish the peer review history of their article (what does this mean?). If published, this will include your full peer review and any attached files.

Reviewer #1: No

Reviewer #2: **Yes: **Vern Carruthers

Reviewer #3: No

---

## [Editor Report · Decision Letter 1]

6 Apr 2022

Dear Dr Turkewitz,

We are pleased to inform you that your manuscript entitled "A novel membrane complex is required for docking and regulated exocytosis of lysosome-related organelles in Tetrahymena thermophila" has been editorially accepted for publication in PLOS Genetics. Congratulations! In the process of revising your manuscript, you have done a thorough and exemplary job of addressing the comments from the first review. The new data included, particularly in Figure 7, are especially helpful in illustrating mucocyst association of the range of novel components of your newly discovered complex. This is an elegant manuscript and will be of broad interest to the field. In reading the revised manuscript, we noted two very minor redundancies in the text that you will likely want to address prior to publication. The first appears on line 58 in the Author Summary (could be/could have) and the second appears on line 643 (including/concerning). Other than these minors issues, all else appears to be in order - please address them as you prepare your final draft for the production team (the editorial team will not need to re-evaluate).

Yours sincerely,

J. David Castle, Ph.D.

Guest Editor

PLOS Genetics

Gregory P. Copenhaver

Editor-in-Chief

PLOS Genetics

Comments from the reviewers (if applicable):

**Data Deposition**

http://datadryad.org/submit?journalID=pgenetics&manu=PGENETICS-D-21-01256R1

**Press Queries**

---

## [Editor Report · Acceptance letter]

11 May 2022

PGENETICS-D-21-01256R1 

A novel membrane complex is required for docking and regulated exocytosis of lysosome-related organelles in Tetrahymena thermophila 

Dear Dr Turkewitz, 

We are pleased to inform you that your manuscript entitled "A novel membrane complex is required for docking and regulated exocytosis of lysosome-related organelles in Tetrahymena thermophila" has been formally accepted for publication in PLOS Genetics! Your manuscript is now with our production department and you will be notified of the publication date in due course.

With kind regards,

Livia Horvath

PLOS Genetics

On behalf of:
